# Modular addition without black-boxes: Compressing explanations of MLPs that compute numerical integration

## Abstract

The goal of mechanistic interpretability is discovering a simple, low-rank algorithm implemented by models. While we can compress activations into features, compressing nonlinear feature-maps—like MLP layers—is an open problem. In this work, we present the first case study in rigorously compressing nonlinear feature-maps. We work in the classic setting of the modular addition models (Nanda et al., 2023), and target a non-vacuous bound on the behaviour of the ReLU MLP in time linear in the parameter-count of the circuit. To study the ReLU MLP analytically, we use the infinite-width lens, which turns post-activation matrix multiplications into approximate integrals. We discover a novel interpretation of the MLP layer in one-layer transformers implementing the "pizza" algorithm (Zhong et al., 2023): the MLP can be understood as evaluating a quadrature scheme, where each neuron computes the area of a rectangle under the curve of a trigonometric integral identity. Our code is available at `https://tinyurl.com/mod-add-integration`.

## 1 Introduction

Neural networks' ability to generalize suggests they implement simpler, low-rank algorithms despite performing high-dimensional computations (Olah et al., 2020). The field of mechanistic interpretability has made tremendous progress in finding low-rank approximations of activations—for example, discovering interpretable features (Bricken et al., 2023; Cunningham et al., 2023). However, finding low-rank approximations of feature-maps—particularly MLP layers—is still an open problem (Elhage et al., 2022).

Finding low-rank approximations of nonlinear feature-maps becomes particularly important in light of recent work that found evidence that sparse linear features do not fully capture the structure of frontier models (Marks et al., 2024; Engels et al., 2024). Interpretations that lack analyses of feature-maps may not be usable for ambitious applications of mechanistic interpretability, like anomaly detection and worst-case guarantees. The expressivity of MLPs constitutes a large attack surface for perturbing the model, so not compressing feature-maps leaves a lot of free parameters in the interpretation. These free parameters diminish our ability to detect anomalous behaviour, or make strong guarantees.

To illustrate the difficulty of compressing MLPs, consider the following toy model comparing the effective parameter count of deep nonlinear networks and deep linear networks: adding or multiplying $k$ matrices of shape $m \times m$. For linear operations, we need only $m^2$ parameters to completely describe the input-output behaviour, regardless of the depth of network. However, introducing nonlinearities like ReLU between these operations increases the effective parameter count to $km^2$, with the complexity growing exponentially with depth. While deep linear networks can be compressed to shallow networks of equivalent width without loss of expressivity, nonlinear networks resist such compression.

Even in the classical toy setting of modular addition models, the MLP layer is treated as a black box. Nanda et al. (2023) finds sparse features to describe the input and output to the MLP layer, however, they do not tell us how the MLP layer processes the input features to generate the output features.

Figure 1: (left) There are finitely many neurons in the model (indexed by $i$). The function $f_x(\xi_i)$ is ReLU applied to the inputs $x$. The weight of the connection to each output $c$ is $g_c(\xi_i)$ times a neuron-specific output-independent normalization factor $w_i$. (right) Taking the limit as the number of neurons goes to infinity turns the sum over neurons into an integral. Compressing the resulting analytic expression allows us to compress the MLP.

In this work, we present a case study in compressing nonlinear feature-maps. We extend the analysis in Nanda et al. (2023); Zhong et al. (2023); Gromov (2023) to the MLP layer, opening up the final black box remaining in the modular addition models (§2). To demonstrate that compressing the feature-map is essential for reducing free parameters, we use the formal compression metric from Gross et al. (2024). We measure compression by the computational complexity of verifying the interpretation, where proving the same bound with a lower complexity budget would correspond to finding an interpretation with fewer free parameters (§3).

We apply an *infinite-width lens*, wherein we treat densely connected MLPs as finite approximations to an infinite-MLP-width limit (Appendix K). The infinite-width lens permits us to turn post-activation matrix multiplications into approximate integrals and study the remaining operations of the network – including nonlinear operations – analytically. We find a low-rank approximation of the nonlinear function implemented by the MLPs of the "pizza" transformers (Zhong et al., 2023): doubling the frequencies of the input representations, that is, mapping from representations of the form $\cos(\frac{k}{2}(a+b)), \sin(\frac{k}{2}(a+b))$ to $\cos(k(a+b)), \sin(k(a+b))$. Building on surprising patterns in the phase-amplitude representation of the MLP pre-activations and neuron-logit map, we find that the MLP can be understood as implementing a quadrature scheme for integrals of the form

$$\int_{-\pi}^{\pi} \text{ReLU}[\cos(\tfrac{k}{2}(a+b)+\phi)]\cos(kc+2\phi)\,\mathrm{d}\phi = \frac{2}{3}\cos(k(a+b-c)),$$

for a handful of key frequencies $k$ where the integral can be seen as the limiting case of the MLP's summation as the number of neurons increases (§4).

We confirm this interpretation by creating non-vacuous bounds for the outputs of these MLPs in time *linear* in the number of parameters, i.e. without evaluating it on all $P^2$ possible inputs (§5). Finally, we resolve the puzzling observation that the logits of supposedly "pizza" models are closer of those of Nanda et al. (2023)'s "clock" algorithm, by explaining how the model uses secondary frequencies equal to twice of each key frequency in order to compensate for how the pizza algorithm fails in cases when $k(a-b) \approx \pi$ (§6).

## 2 BACKGROUND

### 2.1 MECHANISTIC INTERPRETABILITY OF MODULAR ADDITION MODELS

Models trained on the modular addition task have become a classic testbed in the mechanistic interpretability literature. Originally, Nanda et al. (2023) studied a one-layer transformer model. They found low-rank features to describe all components of the model, and analysed the feature-map of the final linear layer. While they generated a human-intuitive algorithm of how the model works, they did not explain *how* the MLP layers compute logits that fit with the form required by this algorithm. Despite the tremendous progress in analysing the non-MLP layers of the model, the ReLU MLP is still treated as a black-box.

Zhong et al. (2023) extended the analysis to a family of architectures parameterized by *attention rate*. The architecture from Nanda et al. (2023) corresponds to attention rate 0, while attention rate

1 corresponds to a ReLU MLP-only model, i.e. a transformer with constant attention. Depending on attention rate, they showed that models may learn the "clock" or "pizza" algorithm. They exhaustively enumerated inputs to MLP-only model and found a description of the feature-map. However, exhaustive enumeration is not feasible for larger input sizes, and does not constitute an insight-rich explanation. Thus, their approximation of the feature-map is equal to the feature-map itself, failing to provide any compression.

Gromov (2023) considered a cleaner version of the MLP-only architecture considered by Zhong et al. (2023), and used quadratic activations instead of ReLU activations. They presented a formula corresponding to a compressed feature-map. However, they did not present sufficient evidence to show that the trained models follow the formula as suggested. They only showed that the weights are roughly "single-frequency", i.e. they are well approximated by the largest Fourier component. Establishing that the model in fact uses the stated algorithm as opposed to a different algorithm would require significantly more validation.

In this work, we analyze the ReLU MLP with the goal of demonstrating *how* it computes the functions described in prior work, and establish this with rigorous evidence and a formal proof.

## 2.2 EXPERIMENTAL SETUP

We study models which implement the 'pizza' algorithm from Zhong et al. (2023): a one-layer ReLU transformer with four heads and constant attention $= \frac{1}{2}$ for all tokens, trained to compute $M : (a, b, \text{'='}) \mapsto (a + b) \bmod p$. As in Zhong et al. (2023), we take $p = 59$. The model takes input $(a, b, \text{'='})$ encoded as one-hot vectors, and we read off the logits $\text{logit}(a, b, c)$ for all possible values $(\bmod\ p)$ above the final sequence position '=', with the largest logit representing its predicted answer.

Since the attention is constant, the logits of the model given input $a, b$ are calculated as:

$$x_i^{(0)} = W_E t_i + p_i \hspace{4cm} \text{Embedding}$$

$$x^{(1)} = x_2^{(0)} + \frac{1}{2} \sum_{j=1}^{4} W_O^j W_V^j \left( x_a^{(0)} + x_b^{(0)} \right) \hspace{2cm} \text{Post attention residual stream}$$

$$N = \text{ReLU}(W_{\text{in}} x^{(1)}) \hspace{4cm} \text{Post ReLU neuron activations}$$

$$x^{(2)} = x^{(1)} + W_{\text{out}} N$$

$$\text{logits} = W_U x^{(2)} = W_U \left( x^{(1)} + W_{\text{out}} \text{ReLU}(W_{\text{in}} x^{(1)}) \right)$$

As such, the model architecture considered has constant attention $= \frac{1}{2}$ throughout, so the model consists of linear embeddings, followed by ReLU, followed by a further linear layer.

As noted in both Nanda et al. (2023) and Zhong et al. (2023), the contribution from the skip connection around the MLP to the logits is small. Combining this with the fact that the attention is uniform across $a, b$, and using $W_L = W_U W_{\text{out}}$ for the neuron-logit map, the logits can approximately be written as the sum of the contributions of each of the $d_{\text{mlp}}$ neurons:

$$\text{logit}(a, b, c) = \sum_{i=1}^{d_{\text{mlp}}} (W_L)_{ci} \cdot \text{ReLU}(\tfrac{1}{2}\text{OV}(a)_i + \tfrac{1}{2}\text{OV}(b)_i)$$

## 2.3 THE "PIZZA" ALGORITHM

Zhong et al. (2023) demonstrated that small transformers using constant attention implement modular arithmetic using the following algorithm, which they call the "pizza" algorithm:

1. $OV(a), OV(b)$ embed the one-hot encoded tokens $a, b$ as $(\cos(ka), \sin(ka))$ and $(\cos(kb), \sin(kb))$ for a small handful of key frequencies $k$

2. Before the MLP layer, the representation of the two tokens are averaged:

$$(\cos(ka) + \cos(ka), \sin(ka) + \sin(ka))/2 = \cos(k(a - b)/2)(\cos(k(a + b)/2), \sin(k(a + b)/2))$$

3. The network then uses the MLP layer to "double the frequencies":

$$N = |\cos(k(a - b)/2)| \, (\cos(k(a + b)), \sin(k(a + b)))$$

4. Finally, $W_L$ scores possible outputs by taking the dot product with $(\cos(kc), \sin(kc))$:

$$\text{logit}(a, b, c) = |\cos(k(a-b)/2)| \left(\cos(k(a+b))\cos(kc) + \sin(k(a+b))\sin(kc)\right)$$
$$= |\cos(k(a-b)/2)|\cos(k(a+b-c))$$

They distinguish this from the "clock" algorithm of Nanda et al. (2023), whose logits have no dependence on $|\cos(k(a-b)/2)|$, and where the MLP layer instead multiplies together its inputs.

Note that Zhong et al. (2023) check that the MLP layer doubles frequencies *empirically* (i.e. by brute force), by treating the MLP as a black box and enumerating the MLP's outputs for all possible inputs. In this work, we seek to understand *how* the MLP performs its role.

## 3 COMPRESSING MLPs

Post-hoc mechanistic interpretability (Olah et al., 2020; Elhage et al., 2021; Black et al., 2022) can be formalized as finding a compact explanation (Gross et al., 2024) of how the model computes its outputs on the entire input distribution. Gross et al. (2024) demonstrated that finding non-trivial compact explanations, i.e. proving a meaningful bound instead of a null bound, necessarily requires more mechanistic information about the model behaviour. Moreover, if some marginal mechanistic analysis does not result in compression, then it does not reduce the free parameters of that component.

Gross et al. (2024) conduct a limited empirical study of models trained to compute the maximum of $k$ integers, which are attention-only transformers with no MLP layers. Our work can be seen as applying this rigorous formalization to a more challenging case study.

Formally, we consider the computational complexity needed to check the behaviour of the MLP on all inputs. The lower the complexity is, the better we have understood the MLP and the better we can compress our description of how the MLP computes the function that it does.

The naive baseline (as provided by both Nanda et al. (2023) and Zhong et al. (2023)) is to describe the MLP's behaviour by evaluating it on every possible input. For $n$ such inputs and an MLP with width $d_{\text{mlp}}$ and input/output dimension $d_{\text{model}}$, this requires checking the result of $\mathcal{O}(n\, d_{\text{mlp}} d_{\text{model}})$ operations. This corresponds to a null interpretation providing zero understanding of how the MLP internally functions.

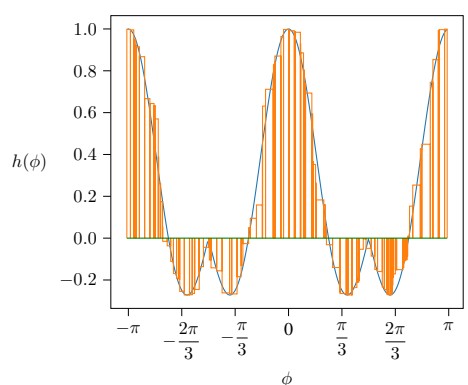

Figure 2: We argue that the MLP approximately computes integrals. We plot the computed integral for frequency $k = 12$ when $a + b = c = 0$. The widths and heights of rectangles are generated by the actual weights in a trained model.

Ideally, we hope that we can bound the MLP's behaviour by referencing only the model weights, without evaluating it on all inputs – in time $\mathcal{O}(d_{\text{mlp}} d_{\text{model}} + n)$, linear in the parameter count of the MLP. As this is the information-theoretic limit, we target interpretations that formally bound MLP behaviour in time linear in parameter count.

## 4 INTERPRETING "PIZZA" MLPs AS PERFORMING NUMERICAL INTEGRATION

Following Nanda et al. (2023), we focus on a particular ("mainline") model in the main body of the work. We confirm that our results generalize to another 150 transformers in Appendix C. First, we identify new structure in the model by using the amplitude-phase form of the Fourier series. Then we describe how to leverage this structure to explain how the MLP approximates an integral to double the frequency of the preactivations.

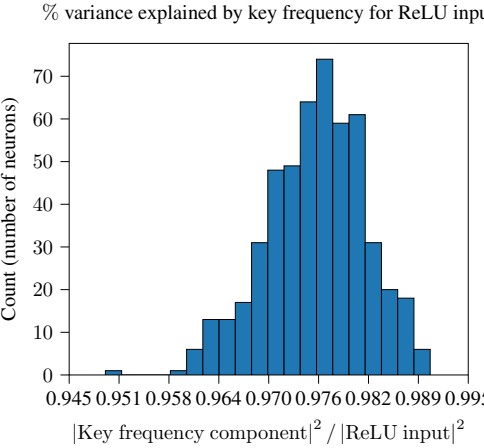

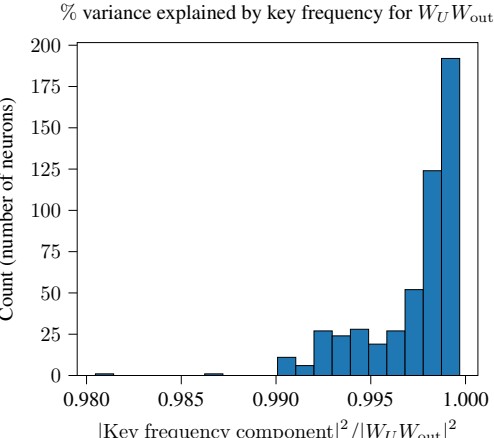

(a) For most neurons, above 96% of the variance of the ReLU input is explained by the largest Fourier frequency component. We compute the square of the largest Fourier coefficient divided by the sum of the squares of all Fourier coefficients and the square of the residual contributions.

(b) For most neurons, above 98% of the variance of the matrix $W_{out}$ is explained by the largest Fourier frequency component. We compute the square of the largest Fourier coefficient divided by the sum of the squares of all Fourier coefficients.

Figure 4: Histograms of the variance explained by the largest Fourier frequency component for the pre-activations and neuron-logit map $W_L$ for each of the 512 neurons in the mainline model.

## 4.1 STUDYING THE MODEL IN THE AMPLITUDE-PHASE FOURIER FORM

We first note that $OV(a), OV(b)$ do not actually embed all inputs to sine and cosine representations of equal magnitude. In general, for each neuron $i$, taking $s = k(a+b)/2$ and $s' = k(a-b)/2$, we can write the representation of the the neuron-$i$ preactivation for an input as $\cos s'(\alpha_i \cos s + \beta_i \sin s)$ for coefficients $\alpha_i, \beta_i$. If we rewrite this expression into the amplitude-phase form of the Fourier series, by putting $(\alpha_i, \beta_i)$ into polar coordinates, our preactivation expression becomes $r_i \cos s'(\cos \phi_i \cos s - \sin \phi_i \sin s) = r_i \cos s' \cos(s + \phi_i)$.

Similarly, $W_L$ does not take the dot product with $(\cos t, \sin t)$ but actually multiplies by $\alpha_i' \cos t + \beta_i' \sin t$. Taking $t = kc$, we can again put this expression into the amplitude-phase Fourier form by putting $(\alpha_i', \beta_i')$ into polar coordinates, giving $r_i'(\cos \psi_i \cos t - \sin \psi_i \sin t) = r_i' \cos(t + \psi_i)$.

The contribution to the logits from neuron $i$ is then $r_i' r_i \operatorname{ReLU}[\cos s' \cos(s + \phi_i)] \cos(t + \psi_i)$.

We find that **each neuron has a primary frequency for both input and output**. As seen in Figure 4, the majority of pre-activations for each neuron consist of terms from a single key frequency, as does the neuron-logit map $W_L$. The largest frequency component for the ReLU pre-activation is almost always *equal* to the largest frequency component for the corresponding row of $W_L$. This allows us to divide the neurons into clusters $I_k$, each of which correspond to a single frequency $k$.

Furthermore, the **output phase is double the input phase**. In Figure 3, we plot the input and

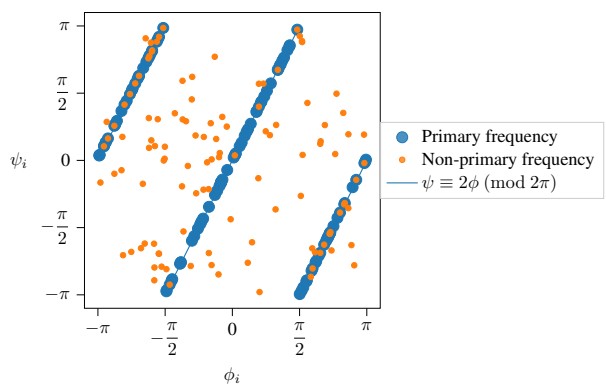

Figure 3: We plot the input and output phase shift angles for frequency $k = 12$, where $\psi_i \approx 2\phi_i \pmod{2\pi}$ for the primary frequency of each neuron. The line has $R^2 = 0.9999$ and the intervals between angles have mean width 0.054, and standard deviation 0.049. This shows that the angles are roughly uniform.

output phase shift angles $\phi_i$ and $\psi_i$ for the largest two frequencies. For each neuron's primary frequency, the neuron's output phase shift $\psi_i$ is almost exactly twice its input phase shift $\phi_i$.

## 4.2 FREQUENCY DOUBLING USING A TRIGONOMETRIC INTEGRAL IDENTITY

Treating the MLP layer as approximately performing numerical integration allows us to make sense of the previous observations. Recall that we can write the logits of the model as

$$\text{logit}(a, b, c) \approx \sum_{i=1}^{d_{\text{mlp}}} (W_L)_{ci} \cdot \text{ReLU}\left(\tfrac{1}{2}\text{OV}(a)_i + \tfrac{1}{2}\text{OV}(b)_i\right)$$
$$\approx \sum_{k \in \mathcal{K}} \sum_{i \in I_k} \cos\left(k(c + 2\phi_i)\right) \text{ReLU}\left(\cos\left(\tfrac{k}{2}(a - b)\right) \cos\left(\tfrac{k}{2}(a + b + \phi_i)\right)\right)$$

For each frequency $k$, we can write the contributions to the logits from neurons of frequency $k$ as :

$$\text{logit}^{(k)}(a, b, c) = \left|\cos\left(\tfrac{k}{2}(a - b)\right)\right| \sum_{i \in I_k} \cos\left(k(c + 2\phi_i)\right) \text{ReLU}\left(\sigma_k \cos\left(\tfrac{k}{2}(a + b + \phi_i)\right)\right)$$

where $\sigma_k = 1$ if $\left|\cos\left(\tfrac{k}{2}(a - b)\right)\right| \geq 0$ and -1 otherwise.

Ignoring the $|\cos(k(a - b)/2)|$ scaling factor (which does not vary per neuron), we claim that the normalized MLP outputs

$$\sum_{i \in I_k} w_i \text{ReLU}[\sigma_k \cos(\tfrac{k}{2}(a + b) + \phi_i)] \cos(kc + 2\phi_i) \tag{1}$$

can we well-thought of as approximating the integral (see Appendix E):

$$\int_{-\pi}^{\pi} \text{ReLU}[\sigma_k \cos(\tfrac{k}{2}(a + b) + \phi)] \cos(kc + 2\phi) \, \mathrm{d}\phi = \frac{2}{3} \cos(k(a + b - c)). \tag{2}$$

Note that the above integral is valid for both $\sigma_k = -1, 1$.

This gives us the desired form for the output logits:

$$\text{logit}(a, b, c) = \sum_k |\cos(k(a - b)/2)| \cos(k(a + b - c)) \tag{3}$$

# 5 VALIDATION VIA COMPACT GUARANTEES ON MLP PERFORMANCE

As evidence for the usefulness of the numerical integration interpretation, we use it to derive non-vacuous bounds on the output of the MLP *on all inputs* in time linear in the parameters.

## 5.1 COMPUTING NUMERICAL INTEGRATION ERROR

The validation of our interpretation as an integral then becomes a mathematical question of evaluating the efficiency of the quadrature scheme

$$\int_{-\pi}^{\pi} h_{a,b,c}(\phi) \, \mathrm{d}\phi \approx \sum_i w_i' h_{a,b,c}(\phi_i)$$

where $h_{a,b,c}(\phi_i) = \text{ReLU}[\sigma_k \cos(\tfrac{k}{2}(a + b) + \phi_i)] \cos(kc + 2\phi_i)$. The absolute error is given by

$$\varepsilon_f := \left| \int_{-\pi}^{\pi} h_{a,b,c}(\phi) \, \mathrm{d}\phi - \sum_i w_i' h_{a,b,c}(\phi_i) \right| \tag{4}$$

and we can compute relative error by computing (the average value over $a, b, c$ of)

$$\varepsilon_0 := \left| \int_{-\pi}^{\pi} h_{a,b,c}(\phi) \, \mathrm{d}\phi \right| \tag{5}$$

and then dividing Equation 4 by Equation 5 to give the relative error $\varepsilon_r := \varepsilon_f / \varepsilon_0$.

Following Gross et al. (2024), if we can compute a non-vacuous error bound (i.e. a relative error that is strictly less than 1) in time linear in the number of parameters of the computation, we can be

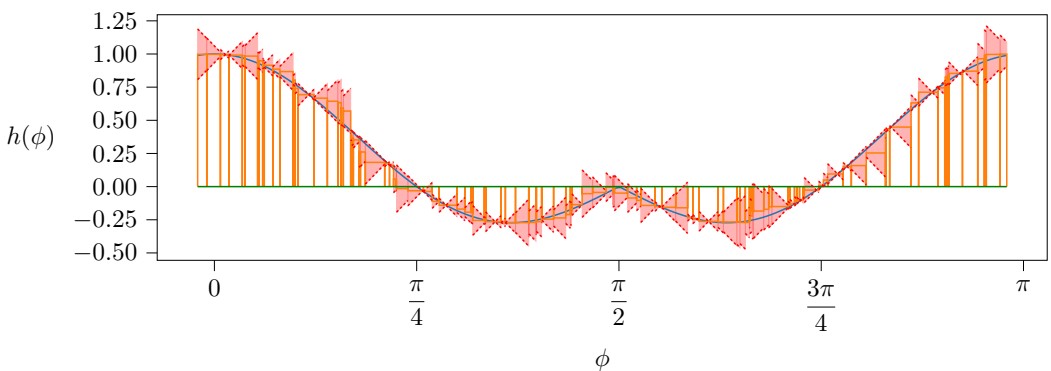

Figure 5: We plot the error bound, which is depicted in red, for frequency $k = 12$. We observe that the red area includes both the actual curve and the numerical integration approximation. The Lipschitz constant is 2, so we bound $h'(\phi)$ by $+2$ and $-2$,

assured that our interpretation has some validity. Since there are $d_{\mathrm{mlp}}$ neurons and $p$ points to evaluate our target is to compute an error bound in time $O(d_{\mathrm{mlp}} + p)$. Thus, this part of the computation is sublinear in the number of parameters as desired.

Recall the neuron contributions to the logits from Equation 2:

$$\mathrm{ReLU}[\sigma_k \cos(\tfrac{k}{2}(a + b) + \phi)] \cos(kc + 2\phi)$$

We can split ReLU as $\mathrm{ReLU}(x) = (x + |x|)/2$. We shall see below that the $x/2$ part integrates to 0. As a result, a relative error bound would not give a meaningful result for this part. However, this part of the network is linear, thus, we can still effectively compress the network behaviour here. For example, we can compute a matrix $A \in \mathbb{R}^{p \times p}$ such that the logit contribution of this part is $A[:, a] + A[:, b]$ for inputs $a$ and $b$. Thus, in the below section, we can restrict our attention to the absolute value part, $|x|/2$.

This turns $\mathrm{ReLU}[\sigma_k \cos(\tfrac{k}{2}(a + b) + \phi)] \cos(kc + 2\phi)$ into

$$\tfrac{1}{2} \underbrace{\left|\sigma_k \cos(\tfrac{k}{2}(a + b) + \phi)\right| \cos(kc + 2\phi)}_{h_{a+b,c,\sigma_k}} + \tfrac{1}{2}\sigma_k \cos(\tfrac{k}{2}(a + b) + \phi) \cos(kc + 2\phi)$$

We sort the phases $\phi_i$ for each neuron, and turn the weights $w_i$ into widths of the rectangles, and the function calls $h(\phi_i)$ into the heights of the rectangles corresponding to the function evaluated at $\phi_i$. This gives a picture similar to Figure 2. The absolute error is

$$\varepsilon_h := \left| \int_{-\pi}^{\pi} h(x) - h(\phi_i) \, \mathrm{d}x \right|. \tag{6}$$

A crude bound is:

$$\leq \int_{-\pi}^{\pi} |h(x) - h(\phi_i)| \, \mathrm{d}x \tag{7}$$

$$\leq \int_{-\pi}^{\pi} |x - \phi_i| \cdot \sup_x |h'(x)| \, \mathrm{d}x \tag{8}$$

$$\leq \sup_x |h'(x)| \sum_i \left( \int_{v_{i-1} - \phi_i}^{v_i - \phi_i} |x| \, \mathrm{d}x \right) \tag{9}$$

$$= \sup_x |h'(x)| \sum_i \frac{1}{2} \begin{cases} \left|(v_i - \phi_i)^2 - (v_{i-1} - \phi_i)^2\right| & \text{if } \phi_i \in [v_{i-1}, v_i] \\ (v_i - \phi_i)^2 + (v_{i-1} - \phi_i)^2 & \text{otherwise} \end{cases} \tag{10}$$

where the rectangle width goes from $v_{i-1}$ to $v_i$ and the function is evaluated at $\phi_i$.

For $h = h_{a+b,c,\sigma_k}$, we can bound

$$\sup_x |h'(x)| \leq 2$$

analytically, and the remaining sum can be evaluated easily in $\mathcal{O}(d_{\mathrm{mlp}})$ time; call this $\varepsilon_{\approx \int}$.

Notice that this upper bound has no dependency on $h$, and therefore is a valid upper bound, for all possible $a, b, c$ triplets. In other words, we can bound the error of the quadrature approximation for any $a, b, c$ by evaluating an expression that takes time linear in the number of neurons.

This bound is represented visually in Figure 5. Note that the figure is produced by analysing the trained model weights.

We must also include approximation error from $\psi_i \approx 2\phi_i$, which we call the "angle approximation error," $\varepsilon_\phi$: we have $\sum_i w_j' |\cos(k_i(a+b)/2 - \phi_i)| \cdot (\cos(kc + \psi_i) - \cos(kc + 2\phi_i)) \leq \sum_i w_j' |\cos(k_i(a+b)/2 - \phi_i)| \cdot |\psi_i - 2\phi_i| \leq \sum_i w_j' |\psi_i - 2\phi_i|$.

## 5.2 Empirical validation

We now provide empirical validation of our interpretation.

**The network is well-approximated as doing numerical integration.** We can compute the actual error $\varepsilon_\int / \varepsilon_0$ empirically, by evaluating the expression over all possible inputs, giving numbers between 0.03 and 0.05, see Table 1.

**The interpretation gives useful compression because integral explanation gives compact non-vacuous bounds.** Our relative error bounds range from 0.48 to 0.7 (i.e. less than 1), see Table 1.

**The bounds are far away from actual error because we don't completely understand numerical integration.** Intuitively, we'd like to center each box at its corresponding $\phi_i$ and com-

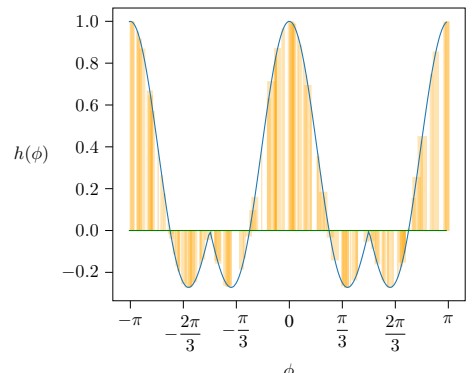

Figure 6: Numerical integration with boxes centered at $\phi_i$ for frequency $k = 12$.

pute the error that results from having box density above or below 1 at various points of the curve (Figure 6). We'd also like to take into account the fact that if one region has a box density above 1 and is adjacent to a region with a box density below one, these density errors *partially* cancel out. However, we don't know how to efficiently compute these effects, and the bounds we're able to give using non-uniform box densities instead of non-uniform angle locations is too crude.

## 6 The role of secondary frequencies

### 6.1 Regressing model logits versus "clock" and "pizza" logits

As noted above, Nanda et al. (2023) claim that logits are of the form ("clock logits")

$$\text{logit}(a, b, c) \propto \cos(k(a + b - c))$$

while Zhong et al. (2023) suggests that logits can also be of the form ("pizza logits")

$$\text{logit}(a, b, c) \propto |\cos(k(a - b)/2)| \cos(k(a + b - c))$$

Interestingly, if we regress the logits against the factors $|\cos(k(a - b)/2)| \cos(k(a + b - c))$, which gives an $R^2$ of 0.86, while if we regress them against just $\cos(k(a + b - c))$, we obtain an $R^2$ of 0.98 – substantially higher. So overall, the "clock logits" give a more accurate expression, suggesting that the analysis above is importantly incomplete. Interestingly, if we only consider the contribution

| Error Bound Equation \ Freq. | 12 | 18 | 21 | 22 |
|---|---|---|---|---|
| $\varepsilon_\int / \varepsilon_0$ (actual error computed by brute force) | 0.05 | 0.03 | 0.05 | 0.03 |
| $(\varepsilon_{\approx\int} + \varepsilon_\phi)/\varepsilon_0$ (error bound computed in linear time) | 0.70 | 0.49 | 0.54 | 0.48 |

Table 1: Relative error bounds by splitting ReLU into absolute value and identity components

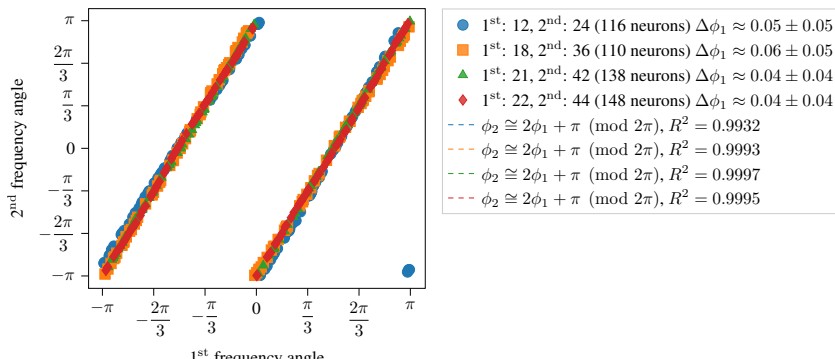

Figure 7: We plot the input phase shifts of the two frequencies. We observe that not only are the secondary frequencies of neurons approximately double the primary frequencies, the input phase shift of the secondary frequencies are approximately twice the primary frequency phase shift plus $\pi$.

to the logits from only the absolute value component of ReLU (Appendix I), the $R^2$ values become 0.99 and 0.85 respectively.

What this suggests is that the absolute value component of ReLU indeed carries out the "pizza" algorithm and produces those logits. As we will demonstrate below, the discrepancy for the overall logits is due to the effects of the substantially smaller non-primary frequencies. In particular, it is explained by the action of the "secondary frequency" – the second largest Fourier component.

### 6.2 USING SECONDARY FREQUENCIES TO BETTER APPROXIMATE CLOCK LOGITS

For each of the neurons, the largest secondary frequency is almost always twice the primary frequency. For example, for neurons of frequency 12, the largest secondary frequency is 24, while for neurons of frequency 22, the largest secondary frequency is 15 ($= 59 - 22 \cdot 2$, note that cosine is symmetric about 0).

Note that the input phase shift of the secondary frequency is approximately twice the input phase shift of the primary frequency plus $\pi$ (Figure 7).

The contribution of the doubled secondary frequency to the logits can thus be written as (compare to Equation 2, note we lose the $\frac{1}{2}$ factor in the pre-ReLU expression because the secondary frequency is double the primary frequency)

$$
\text{logit}^{(2k)}(a,b,c) = \sum_{i \in I_k} \cos{(kc + 2\phi_i)} \, \text{ReLU}\left[\cos{(k(a-b))}\cos{(k(a+b) + 2\phi_i + \pi)}\right]
$$

$$
\approx \int_{-\pi}^{\pi} \cos(kc + 2\phi) \, \text{ReLU}[-\cos{(k(a-b))}\cos(k(a+b) + 2\phi)] \, d\phi
$$

Letting $\theta = \phi + k(a+b)/2$, we get

$$
= \int_{k(a+b)/2 - \pi}^{k(a+b)/2 + \pi} \cos(k(c - (a+b)) + 2\theta) \, \text{ReLU}[-\cos{(k(a-b))}\cos(2\theta)] \, d\theta
$$

Because the integrand is $2\pi$-periodic, the shift on the limits of integration is irrelevant:

$$
= \int_{-\pi}^{\pi} \cos(k(c - (a+b)) + 2\theta) \, \text{ReLU}[-\cos{(k(a-b))}\cos(2\theta)] \, d\theta
$$

By the law of cosine addition:

$$
= \int_{-\pi}^{\pi} \Big[ \cos(k(c - (a+b)))\cos(2\theta) - \sin(k(c - (a+b)))\sin(2\theta) \Big]
$$
$$
\cdot \text{ReLU}[-\cos{(k(a-b))}\cos(2\theta)] \, d\theta
$$

Because $\sin$ is odd and $\text{ReLU}[\cos]$ is even, the $\sin$ term integrates to 0.

$$= \cos(k(c - (a + b))) \int_{-\pi}^{\pi} \cos(2\theta) \, \text{ReLU}[-\cos(k(a - b))\cos(2\theta)] \, d\theta$$

$$= -\tfrac{\pi}{2} \cos(k(a - b)) \cos(k(a + b - c))$$

This logit contribution helps make the model more robust. Note that in the expression for "pizza logits" (Equation 3), the model works by having that $\cos(k(a + b - c))$ is largest when $c = a + b \pmod{p}$, and choosing the largest logit as output. However, this logit difference has a factor of $\left|\cos\left(\frac{k}{2}(a - b)\right)\right|$. As a result, when $\left|\cos\left(\frac{k}{2}(a - b)\right)\right| \approx 0$, the logit difference is small and may not distinguish the correct output. However, from the above expression, we have that (by $\cos(2x) = 2\cos^2(x) - 1$) $\cos(k(a - b)) \approx -1$ so this term contributes positively to the correct logit $\cos(k(a + b - c))$. This compensates for the weakness of the pizza logits in cases where $\left|\cos\left(\frac{k}{2}(a - b)\right)\right| \approx 0$.

## 7 DISCUSSION

We provide a first case study in rigorously compressing nonlinear feature-maps. We demonstrate that interpreting feature-maps reveals additional insight about the model mechanism, even in models that the research community assumes that we understand quite well. Our hope is that this work will inspire additional study of feature-maps in mechanistic interpretability.

The key steps in our derivation of efficient bounds were: splitting the input into orthogonal output-relevant and output-irrelevant directions; decomposing the pre-activations as a product of functions of these axes; reindexing the neurons by the output-relevant direction so that the reindexed post-activations depend *only* on the output-irrelevant directions; and compressing independently over the output-relevant direction term and the output-irrelevant direction term. We believe that some of these steps could be adapted to interpret other MLPs, even those where we cannot derive closed-form analytic representations. At the same time, the bounds we derive in Appendix H could be improved to understand how the network is allocating boxes under the curve, which we hope to address with future work.

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

## A   MODEL TRAINING DETAILS

We train 1-layer transformers with constant attention $= \frac{1}{2}$ (equivalently, with $QK$ clamped to 0), as implemented by TransformerLens (Nanda & Bloom, 2022) with the following parameters

$$\text{d\_vocab} = p + 1 = 59 + 1$$
$$\text{n\_ctx} = 2 + 1$$
$$\text{d\_model} = 128$$
$$\text{d\_mlp} = 512$$
$$\text{d\_head} = 32$$
$$\text{n\_heads} = 4$$
$$\text{n\_layers} = 1$$
$$\text{act\_fn} = \text{'relu'}$$

We take $80\%$ of all $(a, b)$ pairs $\bmod p$ as the training set, and the rest as the validation set. We set the loss function to be the mean log probabilities of the correct logit positions, and train for 10000 epochs using the AdamW optimizer with the default weight_decay $= 0.01$. The large number of epochs is such that the resulting model achieves $100\%$ accuracy on all possible input pairs. We chose 151 random seeds which are pseudorandomly deterministically derived 0, along with 150 values read from `/dev/urandom`.

Each training run takes approximately 11–12 GPU-minutes to complete on an NVIDIA GeForce RTX 3090. In total, the experiments in this paper took less than 1000 GPU-hours.

## B   MORE FIGURES AND RESULTS FOR MAINLINE MODEL

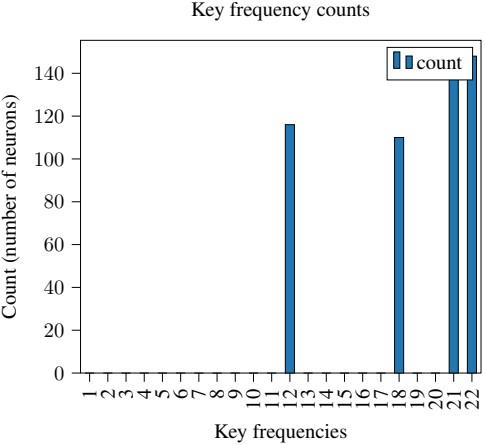

Figure 8: The key frequencies for this model are 12, 18, 21, and 22 (multiplied by $2\pi/p$)

## C   RESULTS FOR OTHER RANDOM SEEDS

To ensure that this phenomenon is not specific to the model we looked at, we trained 151 models with the same setup and different random seeds (leading to different weight initialization and train-test split). We replicate some of the analysis above to show that a significant proportion of these models follow the above explanation.

For our **second observation**, we notice that most neurons are single frequency, with the largest frequency explaining more than 90% of the variance of the ReLU input, see Figure 13. The same is true for the $W_{out}$ matrix, see Figure 14. For our **third observation**, we check that (disregarding neurons where the largest frequency is the constant term) 100 out of 151 models ('good models')

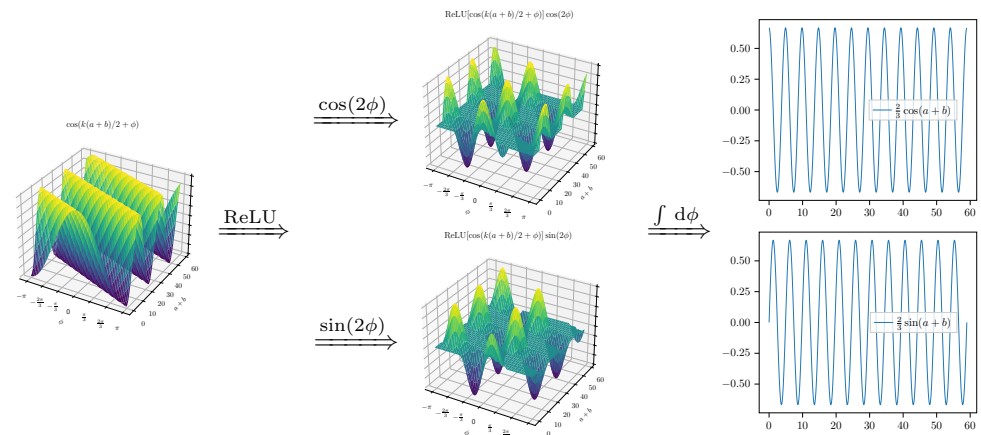

Figure 9: The network computes logits by approximate numerical integration.

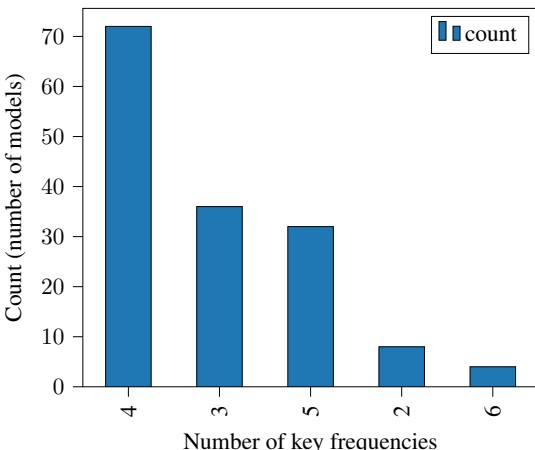

Figure 10: Most models have 3 to 5 key frequencies, which is in line with what we would expect to make the above argument work.

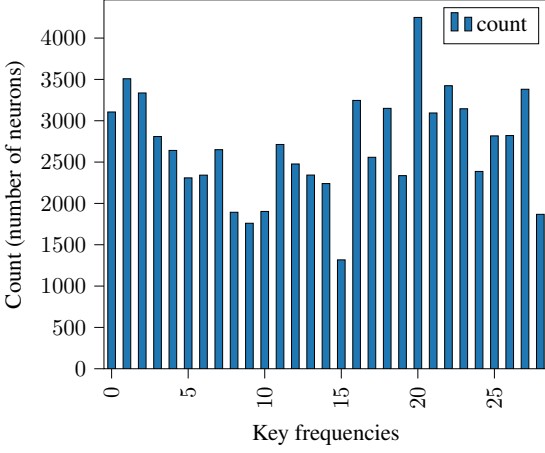

Figure 11: Key frequencies for the models are roughly uniformly distributed across all possible values.

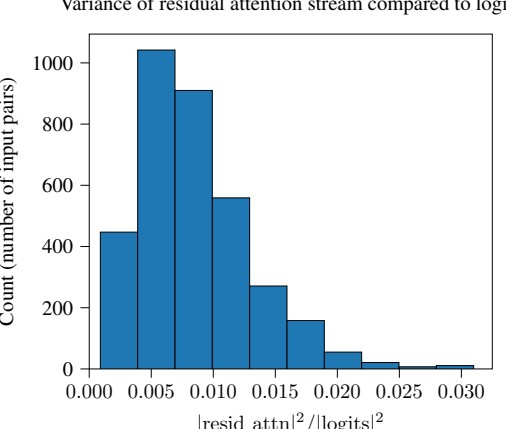

Figure 12: Residual connection from the attention stream causes mostly less than 3% of the variance of the logits.

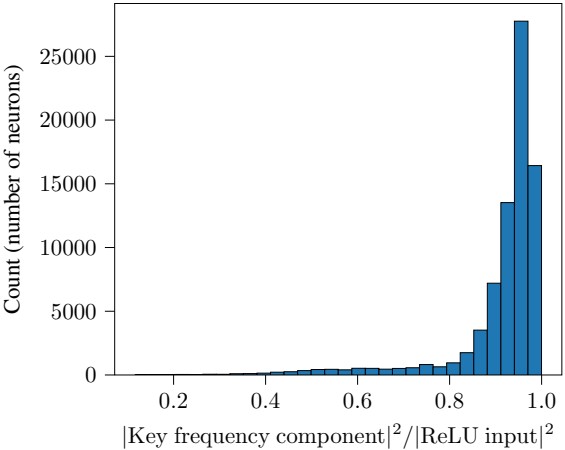

Figure 13: For most neurons, above 90% of the variance of the ReLU input is explained by the largest Fourier frequency component.

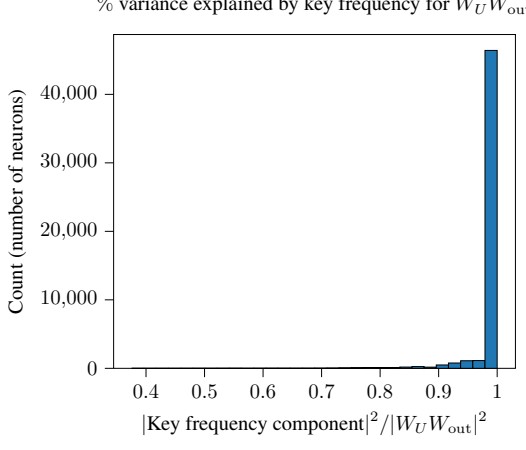

Figure 14: For most neurons in 'good' models, above 98% of the variance of the $W_{out}$ matrix is explained by the largest Fourier frequency component.

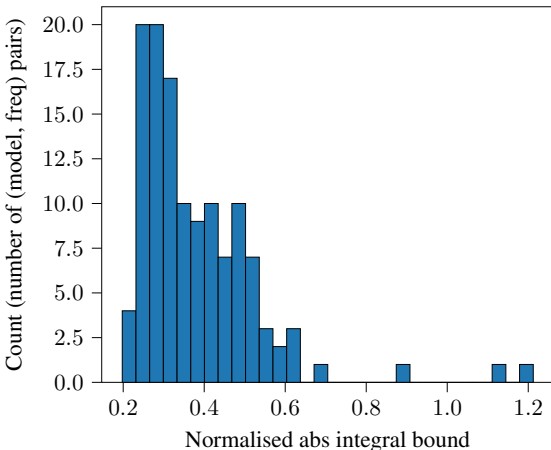

Figure 15: Most of the pairs have normalised abs integral bounds less than $0.6$, compared to the naive abs error bound of $0.85$.

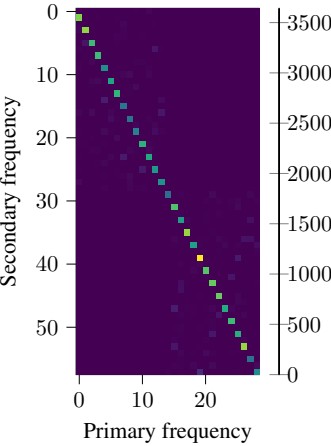

Figure 16: The second largest non-zero Fourier component is 2 times the primary frequency for $83\%$ of the neurons.

have the frequencies matching for all neurons, with a further 27 models where the frequencies don't match for less than 10 (out of 512 neurons). For our **fourth observation**, we look at the 100 good models and see that for 78 of them, we have $R^2 > 0.9$ between the angles for all key frequencies. For our **fifth observation**, we look at the 100 good models and see that for 54 of them, we have *mean width of intervals > standard deviation of width*, showing that angles are roughly uniformly distributed.

Finally, we carry out the error bound calculations for the 100 good models. For each (model, freq) pair, we can calculate the error bounds as in Table 1. We see that for 295 out of 358 such pairs ($82\%$), the empirical errors (normalised abs cos error, normalised abs sin error, normalised id cos error, normalised id sin error) are all less than $0.1$. For these pairs, the normalised abs integral bound has a median of $0.40$, which is $47\%$ of the naive abs bound of $0.85$. Also, $99\%$ of the normalised abs integral bounds are less than the naive abs bound (see Figure 15. Hence, the numerical integration phenomenon explained above indeed appears in most trained models, and our method of proof is able to produce a non-vacuous error bound whenever this phenomenon occurs.

For the relationship observed between primary and secondary frequencies, we see that it still holds when we use different random seeds. In particular, around $84\%$ of the neurons have the second largest non-zero Fourier component being 2 times the primary frequency (see Figure 16). Also, amongst

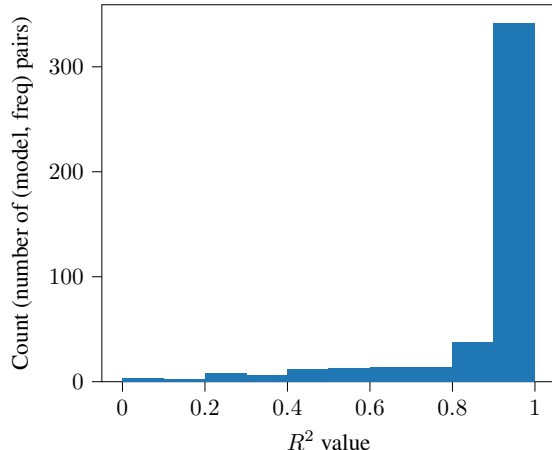

Figure 17: Most of the (model, freq) pairs have the angle for the secondary frequency closely resembling 2 times the primary frequency plus $\pi$.

these neurons, the angle for the secondary frequency is approximate 2 times the angle for the primary frequency $+\pi$, with $R^2 > 0.9$ for most (model, freq) pairs.

## D    TRIGONOMETRIC IDENTITIES

We use the following trigonometric identities throughout the paper:

$$\sin\alpha + \sin\beta = 2\sin\frac{\alpha+\beta}{2}\cos\frac{\alpha-\beta}{2} \tag{11}$$

$$\cos\alpha + \cos\beta = 2\cos\frac{\alpha+\beta}{2}\cos\frac{\alpha-\beta}{2} \tag{12}$$

$$\cos\alpha\cos\beta = \frac{\cos(\alpha+\beta)+\cos(\alpha-\beta)}{2} \tag{13}$$

$$\cos(\alpha+\beta) = \cos\alpha\cos\beta - \sin\alpha\sin\beta \tag{14}$$

$$\cos(\alpha-\beta) = \cos\alpha\cos\beta + \sin\alpha\sin\beta \tag{15}$$

$$\cos(\alpha+\pi) = -\cos\alpha \tag{16}$$

## E    DERIVATION OF TRIG INTEGRAL

Instead of splitting the summation into two chunks, converting each into an integral, and evaluating each integral, we can replace the summand with an integral directly and evaluate that. Also, we use the corrected phase shift $\psi_i = 2\phi_i$:

$$\text{logit}(a,b,c)$$
$$\approx \sum_{k\in\mathcal{K}}\sum_{i\in I_k} w_i\,\text{ReLU}\left(\sigma_k\cos(k\tfrac{a+b}{2}+\phi_i)\right)$$
$$\cos(kc+2\phi_i)$$
$$\approx \sum_{k\in\mathcal{K}} Z_k \int_{-\pi}^{\pi}\text{ReLU}\left(\sigma_k\cos(k\tfrac{a+b}{2}+\phi)\right)$$
$$\cos(kc+2\phi)\,\mathrm{d}\phi$$

Using $\mathcal{F}_{\pm}$ to denote the integral

$$\mathcal{F}_{\pm} = \int_{-\pi}^{\pi}\text{ReLU}\left(\pm\cos(k\frac{a+b}{2}+\phi)\right)\cos(kc+2\phi)\,\mathrm{d}\phi\,,$$

let $s = k\frac{a+b}{2}$ and $t = kc$. Using the periodicity of cosine (Equation 16), we have

$$\mathcal{F}_- = \int_{-\pi}^{\pi} \text{ReLU}\left(-\cos(k\frac{a+b}{2} + \phi)\right)\cos(kc + 2\phi)\,\mathrm{d}\phi$$

$$= \int_{-\pi}^{\pi} \text{ReLU}\left(\cos(k\frac{a+b}{2} + \phi + \pi)\right)\cos(kc + 2\phi + 2\pi)\,\mathrm{d}\phi$$

$$= \int_{0}^{2\pi} \text{ReLU}\left(\cos(k\frac{a+b}{2} + \phi')\right)\cos(kc + 2\phi')\,\mathrm{d}\phi'$$

$$= \int_{-\pi}^{\pi} \text{ReLU}\left(\cos(k\frac{a+b}{2} + \phi)\right)\cos(kc + 2\phi)\,\mathrm{d}\phi$$

$$= \mathcal{F}_+ .$$

So we may write $\mathcal{F} := \mathcal{F}_+ = \mathcal{F}_-$. Note that the integrand is non-zero only when $\phi \in [-\pi/2 - s, \pi/2 - s]$. Applying the cosine product-sum identity (Equation 13) and doing some algebra:

$$\mathcal{F} = \int_{-\pi/2-s}^{\pi/2-s} \cos(s + \phi)\cos(t + 2\phi)\,\mathrm{d}\phi$$

$$= \frac{1}{2}\int_{-\pi/2-s}^{\pi/2-s} \cos(s - t - \phi) + \cos(s + t + 3\phi)\,\mathrm{d}\phi$$

$$= \frac{1}{2}\left[\sin(\phi - s + t) + \frac{1}{3}\sin(s + t + 3\phi)\right]_{-\pi/2-s}^{\pi/2-s}$$

$$= \frac{1}{2}\left[\sin(\pi/2 - 2s + t) + \frac{1}{3}\sin(3\pi/2 - 2s + t)\right]$$

$$- \frac{1}{2}\left[\sin(-\pi/2 - 2s + t) + \frac{1}{3}\sin(-3\pi/2 - 2s + t)\right]$$

Using the periodicity of sine and cosine, we have

$$\mathcal{F} = \frac{1}{2}\left[\sin(\pi/2 - 2s + t) + \frac{1}{3}\sin(3\pi/2 - 2s + t)\right]$$

$$- \frac{1}{2}\left[\sin(-\pi/2 - 2s + t) + \frac{1}{3}\sin(-3\pi/2 - 2s + t)\right]$$

$$= \frac{1}{3}\sin(\pi/2 - 2s + t) + \frac{1}{3}\sin(\pi/2 + 2s - t)$$

$$= \frac{1}{3}\cos(2s - t) + \frac{1}{3}\cos(-2s + t)$$

$$= \frac{2}{3}\cos(2s - t)$$

$$= \frac{2}{3}\cos(k(a + b - c)) ,$$

as desired.

That is, the pizza model computes its logits using the trigonometric integral identity:

$$\int_{-\pi}^{\pi} \text{ReLU}\left(\cos(k(a+b)/2 + \phi)\right)\cos(kc + 2\phi)\,\mathrm{d}\phi$$

$$= \frac{2}{3}\cos(k(a + b - c))$$

Or equivalently:

$$\cos(k(a + b - c))$$

$$= \frac{3}{2}\int_{-\pi}^{\pi} \text{ReLU}\left(\cos(k(a+b)/2 + \phi)\right)\cos(kc + 2\phi)\,\mathrm{d}\phi$$

## F    DERIVATION OF TRIG INTEGRAL INCLUDING SECONDARY FREQUENCIES

Let

$$s_k = k\frac{a+b}{2} \qquad\qquad u_k = k\frac{a-b}{2} \qquad\qquad v_k = \cos(u_k) \qquad\qquad t_k = kc$$

and let $\beta_i$ be the coefficient of the secondary frequency term and $\gamma_i$ be the constant term.

Consider a network of the form

$$\mathrm{logit}(a,b,c) \approx \sum_{k \in \mathcal{K}} \sum_{i \in I_k} w_i \, \mathrm{ReLU}\left[f(ka + \phi_i, kb + \phi_i)\right]\cos(kc + 2\phi_i)$$

for some symmetric even function $f$ which is $2\pi$-periodic – that is, $f(x,y) = f(y,x) = f(-x,y) = f(x,-y) = f(x + 2\pi, y) = f(x, y + 2\pi)$.

Then we have

$$\mathrm{logit}(a,b,c)$$

$$\approx \sum_{k \in \mathcal{K}} \sum_{i \in I_k} w_i \, \mathrm{ReLU}\left[f(ka + \phi_i, kb + \phi_i)\right]\cos(kc + 2\phi_i)$$

$$\approx \sum_{k \in \mathcal{K}} Z_k \int_{-\pi}^{\pi} \mathrm{ReLU}\left[f(ka + \phi, kb + \phi)\right]\cos(kc + 2\phi)\,\mathrm{d}\phi$$

Reindexing with $\theta = \phi + k\frac{a+b}{2} = \phi + s_k$:

$$\approx \sum_{k \in \mathcal{K}} Z_k \int_{k\frac{a+b}{2}-\pi}^{k\frac{a+b}{2}+\pi} \mathrm{ReLU}\left[f(k\tfrac{a-b}{2} + \theta, -k\tfrac{a-b}{2} + \theta)\right]\cos(k(c - (a+b)) + 2\theta)\,\mathrm{d}\theta$$

$$= \sum_{k \in \mathcal{K}} Z_k \int_{s_k-\pi}^{s_k+\pi} \mathrm{ReLU}\left[f(u_k + \theta, -u_k + \theta)\right]\cos(t_k - 2s_k + 2\theta)\,\mathrm{d}\theta$$

Using the fact that the integrand is $2\pi$-periodic and hence we can arbitrarily shift the limits of integration:

$$= \sum_{k \in \mathcal{K}} Z_k \int_{-\pi}^{\pi} \mathrm{ReLU}\left[f(u_k + \theta, -u_k + \theta)\right]\cos(t_k - 2s_k + 2\theta)\,\mathrm{d}\theta$$

Define

$$g_k(\theta) = \mathrm{ReLU}\left[f(u_k + \theta, -u_k + \theta)\right]$$

and note that $g_k$ is even[1] and use the cosine addition formula (Equation 14) to get:

$$= \sum_{k \in \mathcal{K}} Z_k \int_{-\pi}^{\pi} g_k(\theta)(\cos(t_k - 2s_k)\cos(2\theta) - \sin(t_k - 2s_k)\sin(2\theta))\,\mathrm{d}\theta$$

$$= \sum_{k \in \mathcal{K}} Z_k \cos(t_k - 2s_k) \int_{-\pi}^{\pi} g_k(\theta)\cos(2\theta)\,\mathrm{d}\theta - Z_k \sin(t_k - 2s_k) \int_{-\pi}^{\pi} g_k(\theta)\sin(2\theta)\,\mathrm{d}\theta$$

Since $g_k$ is even and $\sin$ is odd, the second integral evaluates to zero, giving

$$= \sum_{k \in \mathcal{K}} Z_k \cos(t_k - 2s_k) \int_{-\pi}^{\pi} g_k(\theta)\cos(2\theta)\,\mathrm{d}\theta$$

---

[1]Because $g_k(-\theta) = \mathrm{ReLU}\left[f(u_k - \theta, -u_k - \theta)\right] = \mathrm{ReLU}\left[f(-(-u_k + \theta), -(u_k + \theta))\right] = \mathrm{ReLU}\left[f(-u_k + \theta, u_k + \theta)\right] = \mathrm{ReLU}\left[f(u_k + \theta, -u_k + \theta)\right] = g_k(\theta)$

In the particular case of secondary frequencies which are double the primary frequencies, with phases also double the primary phases, we have:

$$\text{logit}(a, b, c)$$

$$\approx \sum_{k \in \mathcal{K}} \sum_{i \in I_k} w_i \, \text{ReLU} \left[ \cos(k\tfrac{a-b}{2}) \cos(k\tfrac{a+b}{2} + \phi_i) + \beta_i \cos(k(a-b)) \cos(k(a+b) + 2\phi_i) + \gamma_i \right]$$

$$\cos(kc + 2\phi_i)$$

$$\approx \sum_{k \in \mathcal{K}} Z_k \int_{-\pi}^{\pi} \text{ReLU} \left[ \cos(k\tfrac{a-b}{2}) \cos(k\tfrac{a+b}{2} + \phi) + \bar{\beta}_k \cos(k(a-b)) \cos(k(a+b) + 2\phi) + \bar{\gamma}_k \right]$$

$$\cos(kc + 2\phi) \, \mathrm{d}\phi$$

$$= \sum_{k \in \mathcal{K}} Z_k \int_{-\pi}^{\pi} \text{ReLU} \left[ \cos(u_k) \cos(s_k + \phi) + \bar{\beta}_k \cos(2u_k) \cos(2s_k + 2\phi) + \bar{\gamma}_k \right]$$

$$\cos(t_k + 2\phi) \, \mathrm{d}\phi$$

$$= \sum_{k \in \mathcal{K}} Z_k \int_{-\pi}^{\pi} \text{ReLU} \left[ v_k \cos(s_k + \phi) + \bar{\beta}_k (2v_k^2 - 1) \cos(2s_k + 2\phi) + \bar{\gamma}_k \right]$$

$$\cos(t_k + 2\phi) \, \mathrm{d}\phi$$

Reindexing with $\theta = \phi + s_k$:

$$= \sum_{k \in \mathcal{K}} Z_k \int_{s_k - \pi}^{s_k + \pi} \text{ReLU} \left[ v_k \cos(\theta) + \bar{\beta}_k (2v_k^2 - 1) \cos(2\theta) + \bar{\gamma}_k \right] \cos(t_k - 2s_k + 2\theta) \, \mathrm{d}\theta$$

Using the fact that the integrand is $2\pi$-periodic:

$$= \sum_{k \in \mathcal{K}} Z_k \int_{-\pi}^{\pi} \text{ReLU} \left[ v_k \cos(\theta) + \bar{\beta}_k (2v_k^2 - 1) \cos(2\theta) + \bar{\gamma}_k \right] \cos(t_k - 2s_k + 2\theta) \, \mathrm{d}\theta$$

Define

$$f_k(\theta) = \text{ReLU} \left[ v_k \cos(\theta) + \bar{\beta}_k (2v_k^2 - 1) \cos(2\theta) + \bar{\gamma}_k \right]$$

and use the cosine addition formula to get:

$$= \sum_{k \in \mathcal{K}} Z_k \int_{-\pi}^{\pi} f_k(\theta)(\cos(t_k - 2s_k) \cos(2\theta) - \sin(t_k - 2s_k) \sin(2\theta)) \, \mathrm{d}\theta$$

$$= \sum_{k \in \mathcal{K}} Z_k \cos(t_k - 2s_k) \int_{-\pi}^{\pi} f_k(\theta) \cos(2\theta) \, \mathrm{d}\theta - Z_k \sin(t_k - 2s_k) \int_{-\pi}^{\pi} f_k(\theta) \sin(2\theta) \, \mathrm{d}\theta$$

Since $f_k$ is even and sin is odd, the second integral evaluates to zero, giving

$$= \sum_{k \in \mathcal{K}} Z_k \cos(t_k - 2s_k) \int_{-\pi}^{\pi} f_k(\theta) \cos(2\theta) \, \mathrm{d}\theta$$

# G    NUMERICAL INTEGRATION ERROR BOUND FOR ReLU FUNCTION

Recall that in subsection 5.2, we computed the error bound for integrating the absolute value function rather than the ReLU function in the network. This is because we can break down

$$\text{ReLU}(x) = \frac{x}{2} + \frac{|x|}{2}$$

and the identity part of ReLU integrates to zero. Hence, the baseline result (of approximating the integral to zero) makes more sense when we only consider the absolute value part of ReLU.

However, using the general form of our error bound (Equation 6), it is simple to replicate the analysis for the whole ReLU function. We have the same bound

$$\sup_x |h'(x)| \leq 2$$

| Error Bound Type \ Freq. | 12 | 18 | 21 | 22 | Equation |
|---|---|---|---|---|---|
| Normalised ReLU $\cos$ error | 0.05 | 0.04 | 0.04 | 0.03 | (4)/(6) |
| Normalised ReLU $\sin$ error | 0.03 | 0.05 | 0.04 | 0.03 | (4)/(6) |
| Angle approximation error | 0.13 | 0.07 | 0.06 | 0.05 | (18) |
| Numerical ReLU $\int_{-\pi}^{\pi}$ bound | 0.55 | 0.44 | 0.42 | 0.37 | (19) |
| Numerical ReLU $\int_{0}^{\pi}$ bound | 0.21 | 0.15 | 0.17 | 0.15 | (20) |
| Total numerical ReLU $\int_{-\pi}^{\pi}$ bound | 0.68 | 0.50 | 0.48 | 0.42 | (19) + (18) |
| Total numerical ReLU $\int_{0}^{\pi}$ bound | 0.55 | 0.37 | 0.40 | 0.34 | $2 \cdot (20) + (18)$ |
| Naive ReLU $\cos$ baseline | 0.42 | 0.42 | 0.42 | 0.42 | (5), (17) |
| Naive ReLU $\sin$ baseline | 0.42 | 0.42 | 0.42 | 0.42 | (5), (17) |

Table 2: Error bounds for the ReLU function

Moreover, utilising the fact that ReLU evaluates to 0 on half the range of integration, we can reduce our error bound by only considering the largest half of the boxes (with some boundary effects). This gives us the following error bounds:

Note the baseline is halved because the identity component of ReLU yields a zero integral.

## H  DETAILED COMPUTATION OF NUMERICAL INTEGRATION ERROR

The maximum empirical error (obtained by evaluating the expression for each value of $a + b \mod p$) is shown below:

| Error Bound Type \ Freq. | 12 | 18 | 21 | 22 | Equation |
|---|---|---|---|---|---|
| Normalised abs $\cos$ error | 0.04 | 0.03 | 0.04 | 0.03 | (4)/(6) |
| Normalised abs $\sin$ error | 0.05 | 0.05 | 0.03 | 0.02 | (4)/(6) |
| Normalised id $\cos$ error | 0.06 | 0.05 | 0.04 | 0.04 | (4)/(6) |
| Normalised id $\sin$ error | 0.02 | 0.05 | 0.04 | 0.04 | (4)/(6) |
| Angle approximation error | 0.14 | 0.07 | 0.06 | 0.06 | (18) |
| Numerical abs $\int_{-\pi}^{\pi}$ bound | 0.59 | 0.52 | 0.50 | 0.44 | (19) |
| Numerical abs $\int_{0}^{\pi}$ bound | 0.23 | 0.17 | 0.20 | 0.17 | (20) |
| Total numerical abs $\int_{-\pi}^{\pi}$ bound | 0.73 | 0.59 | 0.56 | 0.50 | (19) + (18) |
| Total numerical abs $\int_{0}^{\pi}$ bound | 0.59 | 0.41 | 0.46 | 0.40 | $2 \cdot (20) + (18)$ |
| Naive abs $\cos$ baseline | 0.85 | 0.85 | 0.85 | 0.85 | (5), (17) |
| Naive abs $\sin$ baseline | 0.85 | 0.85 | 0.85 | 0.85 | (5), (17) |

Table 3: Error bounds by splitting ReLU into absolute value and identity components

The first four rows (normalised abs & id $\cos$ & $\sin$ error) compute the error by brute force exactly: $\left| \int_{-\pi}^{\pi} h(x) - h(\phi_i) \, dx \right|$ for $h \in \{h_{a+b,|C|}, h_{a+b,|S|}, h_{a+b,C}, h_{a+b,S}\}$. Rows ten and eleven compute the baseline for the error $\left| \int_{-\pi}^{\pi} h(x) \, dx \right|$ for $h \in \{h_{a+b,|C|}, h_{a+b,|S|}\}$, which is given by

$$\mathbb{E}_{0 < a+b \le p} \left| \tfrac{4}{3} \cos(2\pi k(a+b)/p) \right| \approx \mathbb{E}_{0 < a+b \le p} \left| \tfrac{4}{3} \sin(2\pi k(a+b)/p) \right| \tag{17}$$

Note that the baseline for $h \in \{h_{a+b,C}, h_{a+b,S}\}$ is 0. Line five (angle discrepancy) computes the error from $\psi \approx 2\phi$:

$$\sum_i w_j' |\psi_i - 2\phi_i|. \tag{18}$$

Line six (numerical abs $\int_{-\pi}^{\pi}$ bound) computes the error bound from the integral:

$$\min_\theta \sum_i \begin{cases} \left| (v_i - (\phi_i - \theta))^2 - (v_{i-1} - (\phi_i - \theta))^2 \right| & \text{if } \phi_i \in [v_{i-1}, v_i] \\ (v_i - (\phi_i - \theta))^2 + (v_{i-1} - (\phi_i - \theta))^2 & \text{otherwise} \end{cases} \tag{19}$$

Shifting by $\theta$ is permitted because our functions are periodic with period $\pi$; it does not matter where we start the integral. Line seven (numerical abs $\int_0^\pi$ bound) takes advantage of the fact that $h$ is $\pi$-periodic to integrate from 0 to $\pi$: taking $\hat{w}_i := w_i'/2$, $\hat{v}_i := v_i/2$, and $\hat{\phi}_i := \phi_i$ for $\phi_i \geq 0$ and $\hat{\phi}_i := \phi_i + \pi$ for $\phi_i < 0$, and we compute

$$\min_\theta \sum_i \begin{cases} \left| (\hat{v}_i - (\hat{\phi}_i - \theta))^2 - (\hat{v}_{i-1} - (\hat{\phi}_i - \theta))^2 \right| & \text{if } \hat{\phi}_i \in [\hat{v}_{i-1}, \hat{v}_i] \\ (\hat{v}_i - (\hat{\phi}_i - \theta))^2 + (\hat{v}_{i-1} - (\hat{\phi}_i - \theta))^2 & \text{otherwise} \end{cases} \tag{20}$$

Line eight (total numerical abs $\int_{-\pi}^\pi$ bound) computes the combined error bound from the integral and angle discrepancy. Line nine (total numerical abs $\int_0^\pi$ bound) takes advantage of the fact that $h$ is $\pi$-periodic to integrate from 0 to $\pi$. This allows us to overlap the two halves of sampled points to try and reduce the error of integration. (In this way, the rectangles in the approximation are narrower and so the error would be smaller.)

Lines six and seven (numerical abs $\int$ bound) also both take advantage of the fact that the function is $2\pi$-periodic, allowing us to shift the intervals formed above by any constant. When bounding approximation error, we use the shift that gives the lowest bound.

The exact error is much smaller than the size of the integral, and the mathematical error bound is also smaller than the size of the integral. This gives convincing evidence that the model is indeed performing numerical integration.

# I ANALYSIS OF THE 'IDENTITY' COMPONENT OF RELU

We can break down ReLU into two parts,

$$\text{ReLU}(x) = \frac{x}{2} + \frac{|x|}{2}$$

The integrals then split into

$$\int_{-\pi}^\pi \cos(-2\phi)\tfrac{1}{2}\cos(\tfrac{k}{2} + \phi)\, \mathrm{d}\phi = \tfrac{2}{3}\cos(k)$$
$$\int_{-\pi}^\pi \sin(-2\phi)\tfrac{1}{2}\cos(\tfrac{k}{2} + \phi)\, \mathrm{d}\phi = \tfrac{2}{3}\sin(k)$$
$$\int_{-\pi}^\pi \cos(-2\phi)\tfrac{1}{2}|\cos(\tfrac{k}{2} + \phi)|\, \mathrm{d}\phi = 0$$
$$\int_{-\pi}^\pi \sin(-2\phi)\tfrac{1}{2}|\cos(\tfrac{k}{2} + \phi)|\, \mathrm{d}\phi = 0$$

We see that the 'identity' part of the ReLU yields a zero integral. So does this part of the model contribute to the logits? It turns out that the answer is yes. To resolve this issue, we look at the discrepancy between the results suggested by previous work: Zhong et al. (2023) claim that logits are of the form

$$\text{logit}(a, b, c) \propto |\cos(k(a - b)/2)| \cos(k(a + b - c))$$

while Nanda et al. (2023) claim that logits are of the form

$$\text{logit}(a, b, c) \propto \cos(k(a + b - c))$$

To check which is correct, we regress the logits against the factors $|\cos(k(a - b)/2)| \cos(k(a+b-c))$, which gives an $R^2$ of 0.86, while if we regress them against just $\cos(k(a + b - c))$, we obtain an $R^2$ of 0.98. So overall, Nanda et al. (2023) give a more accurate expression, but this seems to go against the analysis we did above, which led to the expression in Zhong et al. (2023). (A similar value is obtained if we just use the MLP output and drop the residual streams.) However, if we only consider the contribution to the logits from the absolute value component of ReLU, the $R^2$ values become 0.99 and 0.85 respectively. Therefore, although the contribution from the identity component of ReLU is small, it does make a difference towards reducing the logit dependence on $a - b$, in particular $|\cos(k(a - b)/2)|$. This is a good thing because when $\cos(k(a - b)/2)$ is small, the logit difference between the correct logit $(a + b)$ and other logits will also be small, which will lead to a higher loss. The identity component slightly counters this effect. We can rewrite the identity component as:

$$W_\text{out}\text{OV}(a)/2 + W_\text{out}\text{OV}(b)/2 + W_\text{out}\text{embed}(b)/2$$

Thus, we can store the matrices $\text{logit\_id1}[:, a] = W_{\text{out}}\text{OV}(a)/2$ and $\text{logit\_id2}[:, a] = W_{\text{out}}\text{embed}(a)/2$, then we have

$$F_2(a, b)_c = \text{residual stream} + \text{absolute value terms}$$
$$+ \text{logit\_id1}[c, a] + \text{logit\_id1}[c, b] + \text{logit\_id2}[c, b]$$

We carry out a 2D Fourier transform to find out the decomposition of the $\text{logit\_id1}$ and $\text{logit\_id2}$ matrices (because $W_{\text{out}}$ and $\text{OV}(a)$ are sparse in the (1D) Fourier basis, so their product will naturally be sparse in the 2D Fourier basis). We get $\text{logit\_id1}[c, a] \approx 2\Re(\sum_k a_k e^{i(kc-2ka)})$, where the frequencies $k$ here are the same as subsection 2.3. Hence, the output from the identity component of ReLU is (ignoring $\text{logit\_id2}$ for now, which comes from the residual stream and is smaller): $\sum_k D_k(\cos(kc - 2ka) + \cos(kc - 2kb)) + E_k(\sin(kc - 2ka) + \sin(kc - 2kb)) = \sum_k \cos(k(b - a))(D_k \cos(k(c - a - b)) + E_k \sin(k(c - a - b)))$.

The imaginary component of the FT is very small, $c_k \approx 0$; so the contribution is $\sum_k b_k \cos(k(b - a)) \cos(k(a + b - c))$.

Why does this happen, and why does it help explain the $R^2$ values we got above? We first list the approximate coefficients $a_k$:

| Frequency | 12 | 18 | 21 | 22 |
|---|---|---|---|---|
| abs coefs ($C_k$) | 13.9 | 15.1 | 12.1 | 11.2 |
| id coefs ($D_k$) | -3.7 | -3.9 | -3.2 | -3.3 |

Thus, the overall expression for the logits is

$$F_2(a, b)_c \approx \sum_k (C_k |\cos(k(b - a)/2)|$$
$$+ D_k \cos(k(b - a))) \cos(k(a + b - c))$$
$$= \sum_k (2D_k^2 |\cos(k(b - a)/2)|^2$$
$$+ C_k |\cos(k(b - a)/2)|) \cos(k(a + b - c))$$
$$- D_k \cos(k(a + b - c))$$

using double angle formula. Since $D_k < 0$, the $\sum_k -D_k \cos(k(a + b - c))$ term gives some cushion for the base performance of the model (since as we discussed, the $\cos(k(a + b - c))$ term is why the model gives the highest logit when $c = a + b$). Moreover, the $2D_k^2 |\cos(k(b - a)/2)|^2$ term also further improves the model since it is always non-negative. Hence, the contribution of the identity term evens out parts of the model and improves the logit difference when $|\cos(k(b - a)/2)|$ is small (where the absolute value part doesn't do well). Note that the model would work on its own if we only use the absolute value part, but since ReLU is composed of both the absolute value and identity part and the coefficients combine both parts in a way that improve model performance.

## J  OTHER PLOTS

In this section we display variants of Figure 2, Figure 3, and Figure 5 for the other frequencies.

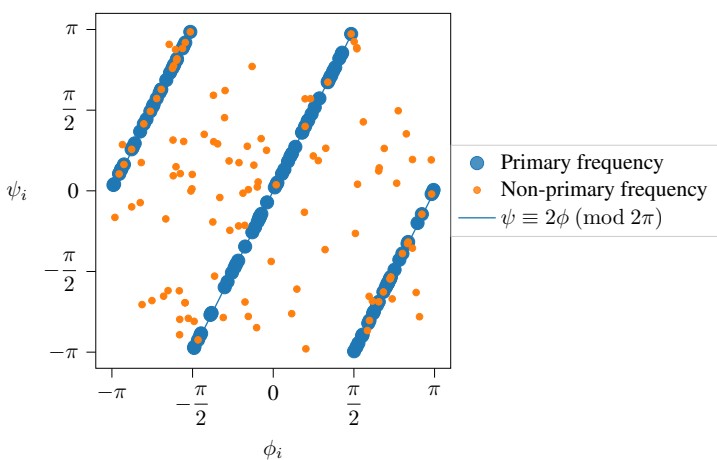

Figure 18: Angles for frequency $k = 12$. $\psi_i \approx 2\phi_i \pmod{2\pi}$ for the primary frequency of each neuron but not in general.

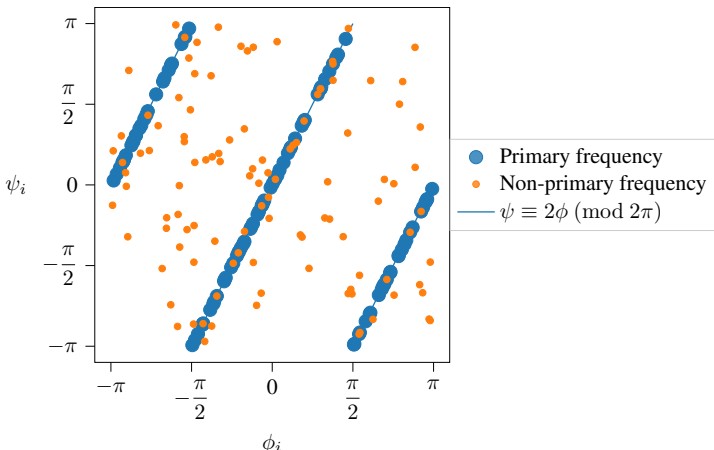

Figure 19: Angles for frequency $k = 18$. $\psi_i \approx 2\phi_i \pmod{2\pi}$ for the primary frequency of each neuron but not in general.

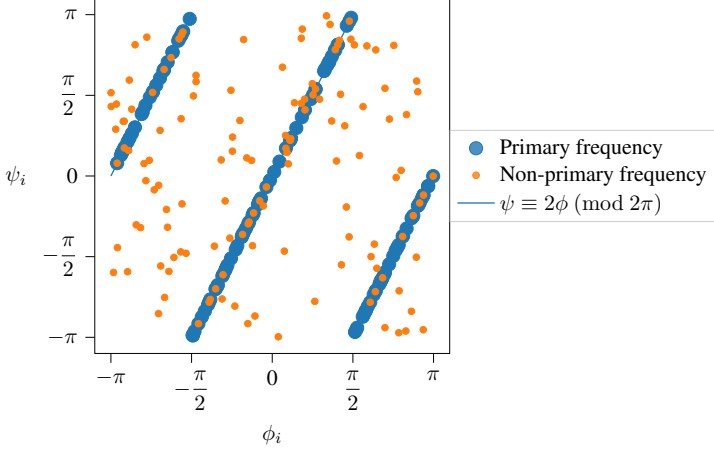

Figure 20: Angles for frequency $k = 21$. $\psi_i \approx 2\phi_i \pmod{2\pi}$ for the primary frequency of each neuron but not in general.

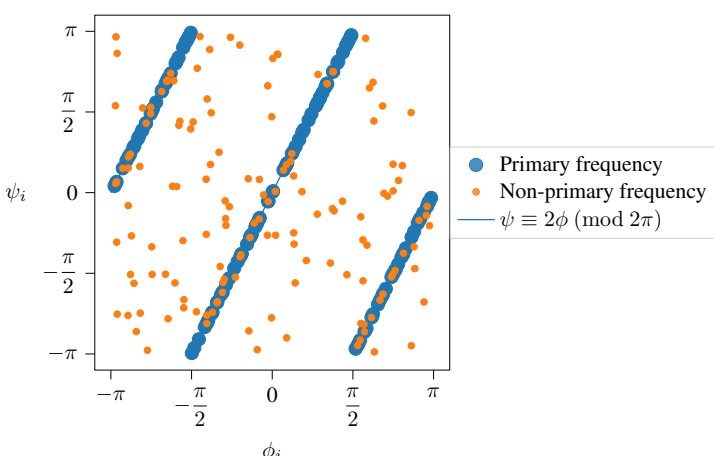

Figure 21: Angles for frequency $k = 22$. $\psi_i \approx 2\phi_i \pmod{2\pi}$ for the primary frequency of each neuron but not in general.

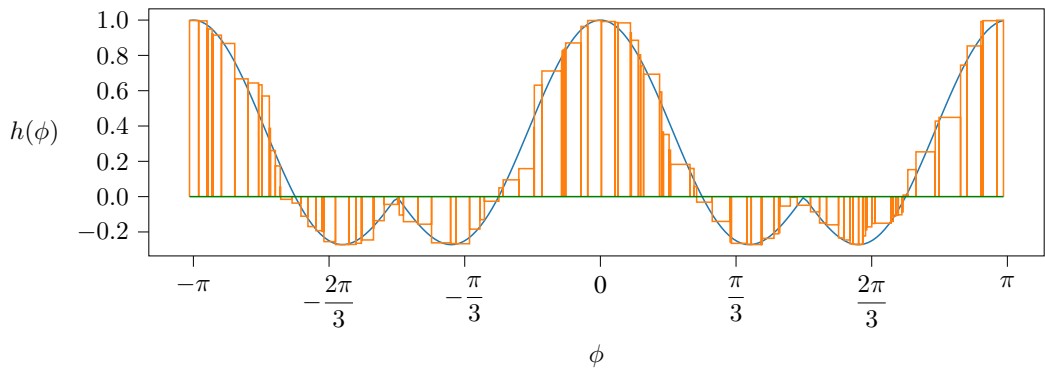

Figure 22: Converting the weighted sum into rectangles to estimate an integral (for frequency $k = 12$). $h(\phi) = |\cos(\phi)| \cos(2\phi)$.

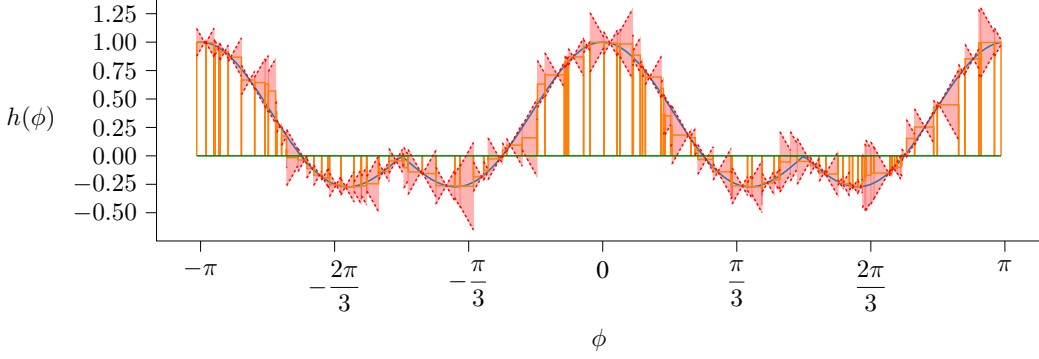

Figure 23: Error bound is the red area (for frequency $k = 12$). Note how the red area includes both the actual curve and the numerical integration approximation.

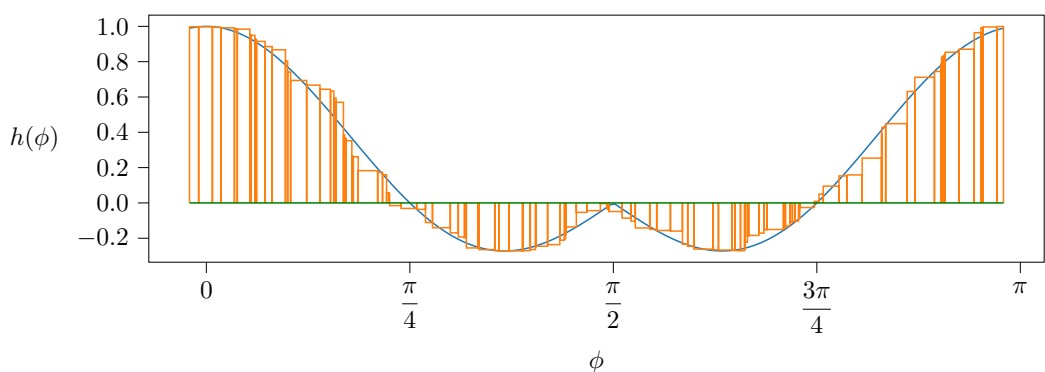

Figure 24: Converting the weighted sum into rectangles to estimate an integral (for frequency $k = 12$). $h(\phi) = |\cos(\phi)| \cos(2\phi)$.

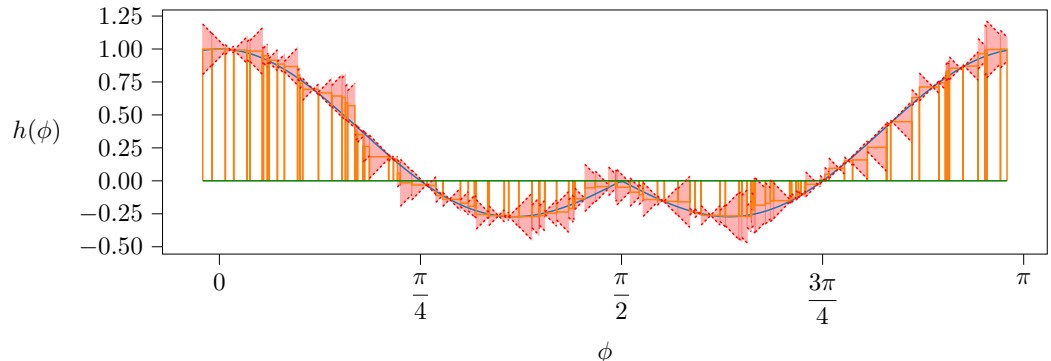

Figure 25: Error bound is the red area (for frequency $k = 12$). Note how the red area includes both the actual curve and the numerical integration approximation.

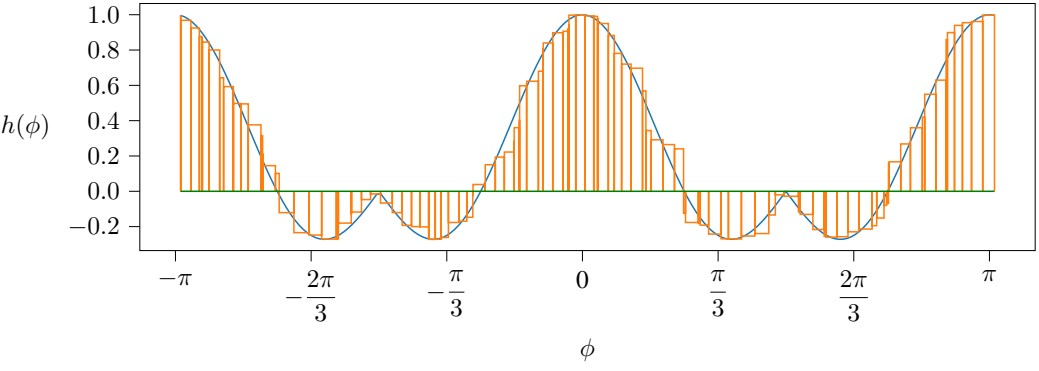

Figure 26: Converting the weighted sum into rectangles to estimate an integral (for frequency $k = 18$). $h(\phi) = |\cos(\phi)| \cos(2\phi)$.

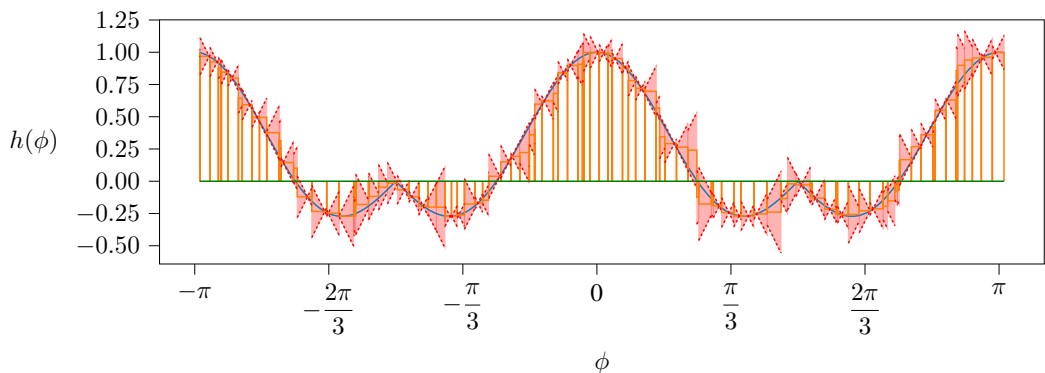

Figure 27: Error bound is the red area (for frequency $k = 18$). Note how the red area includes both the actual curve and the numerical integration approximation.

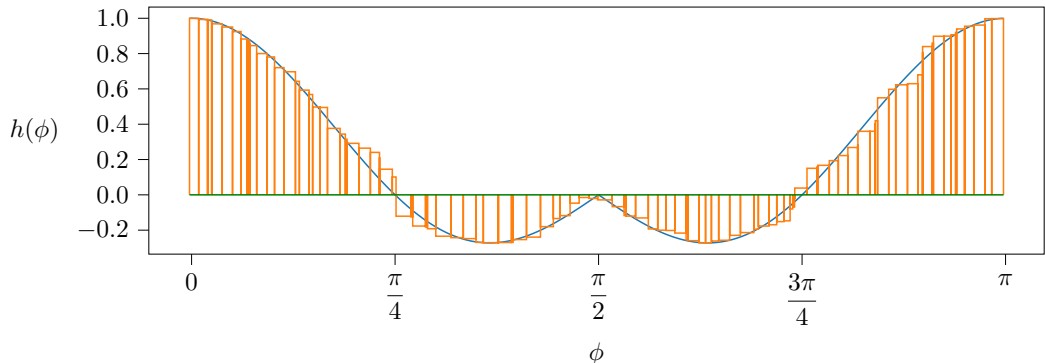

Figure 28: Converting the weighted sum into rectangles to estimate an integral (for frequency $k = 18$). $h(\phi) = |\cos(\phi)| \cos(2\phi)$.

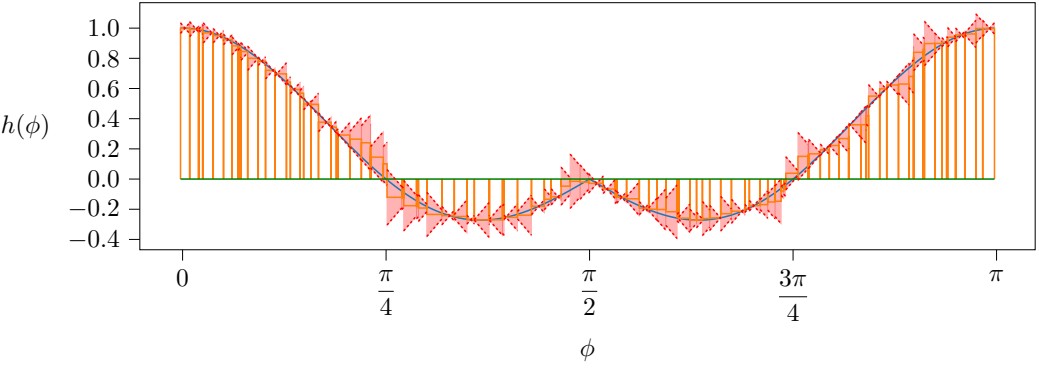

Figure 29: Error bound is the red area (for frequency $k = 18$). Note how the red area includes both the actual curve and the numerical integration approximation.

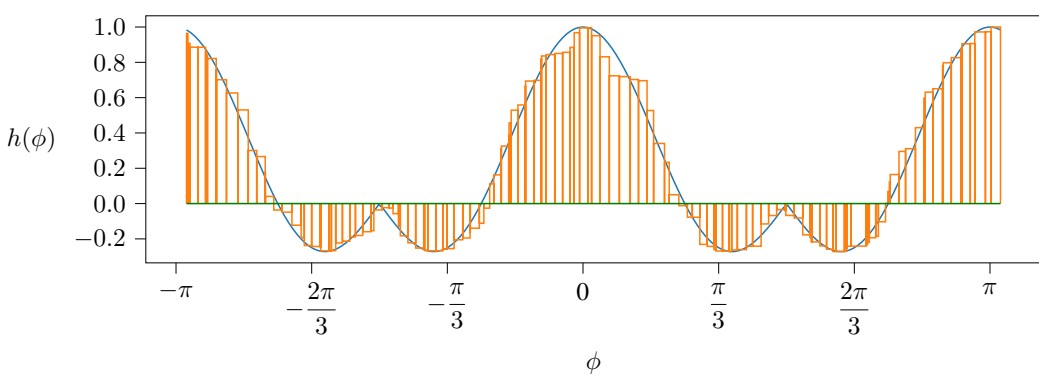

Figure 30: Converting the weighted sum into rectangles to estimate an integral (for frequency $k = 21$). $h(\phi) = |\cos(\phi)| \cos(2\phi)$.

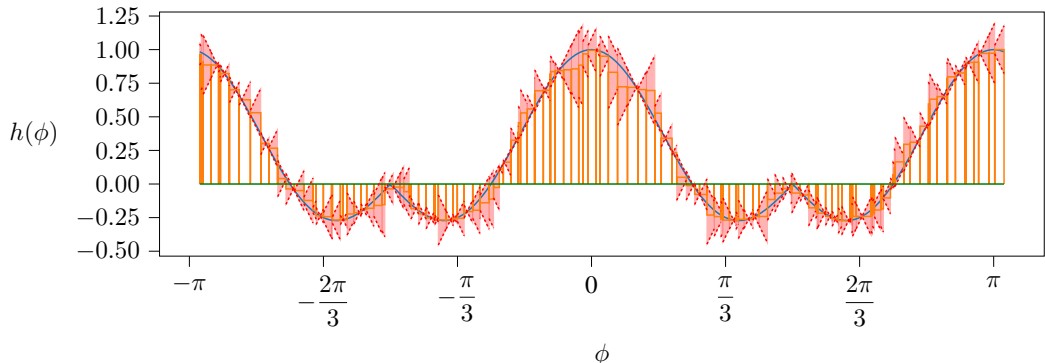

Figure 31: Error bound is the red area (for frequency $k = 21$). Note how the red area includes both the actual curve and the numerical integration approximation.

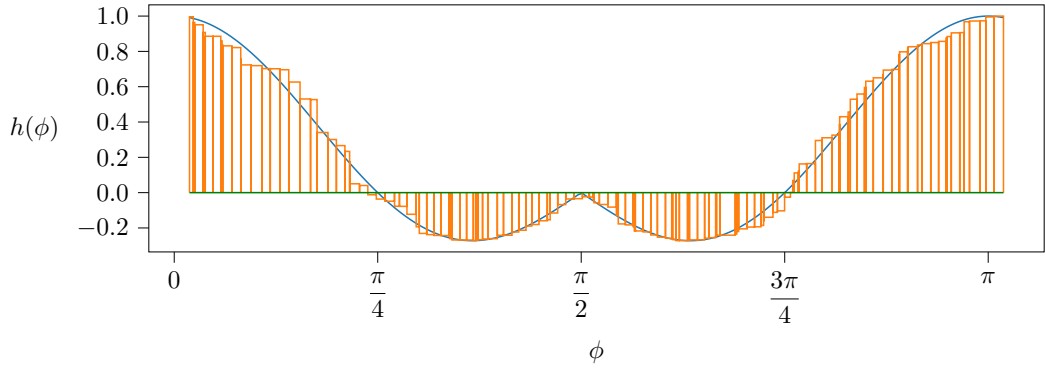

Figure 32: Converting the weighted sum into rectangles to estimate an integral (for frequency $k = 21$). $h(\phi) = |\cos(\phi)| \cos(2\phi)$.

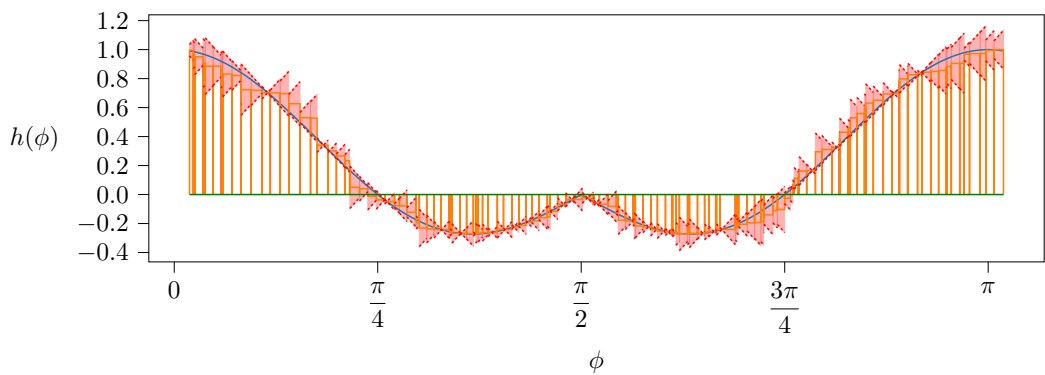

Figure 33: Error bound is the red area (for frequency $k = 21$). Note how the red area includes both the actual curve and the numerical integration approximation.

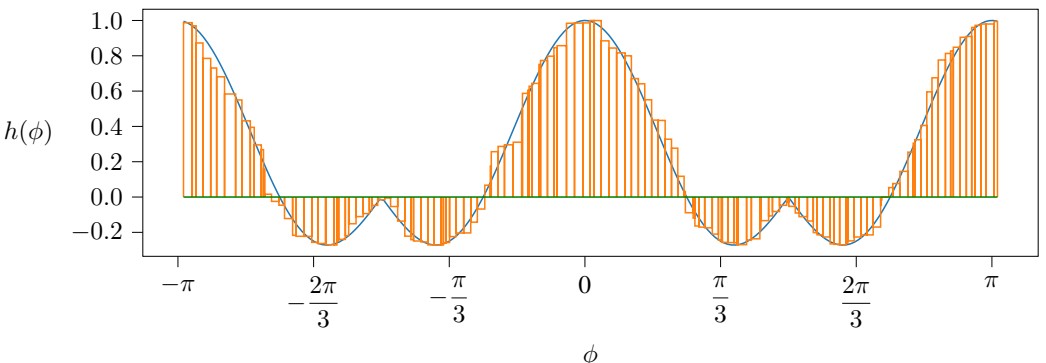

Figure 34: Converting the weighted sum into rectangles to estimate an integral (for frequency $k = 22$). $h(\phi) = |\cos(\phi)| \cos(2\phi)$.

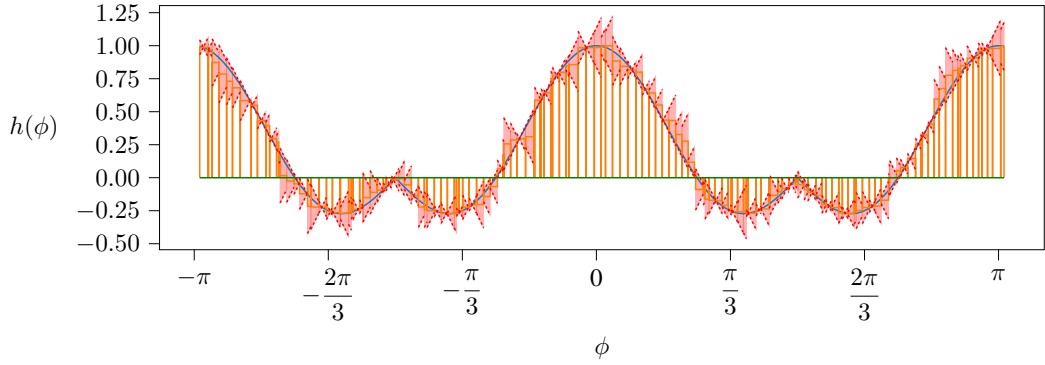

Figure 35: Error bound is the red area (for frequency $k = 22$). Note how the red area includes both the actual curve and the numerical integration approximation.

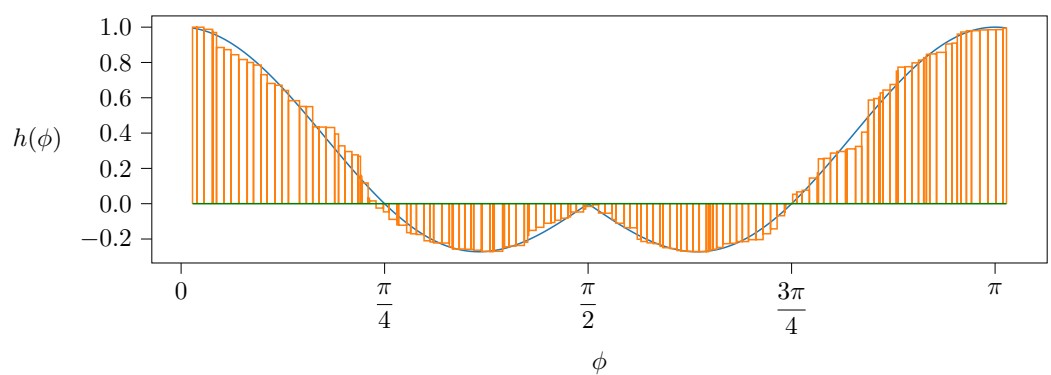

Figure 36: Converting the weighted sum into rectangles to estimate an integral (for frequency $k = 22$). $h(\phi) = |\cos(\phi)| \cos(2\phi)$.

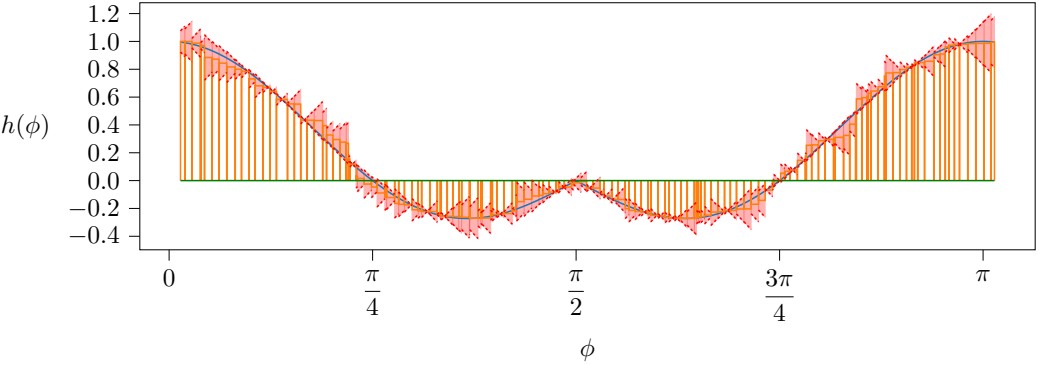

Figure 37: Error bound is the red area (for frequency $k = 22$). Note how the red area includes both the actual curve and the numerical integration approximation.

## K  THE INFINITE-WIDTH LENS

Suppose you have some linear layer:

$$\text{MLP}^{(l)}\left(\vec{a}^{(l-1)}\right) = W_{\text{out}}^{(l)} \text{ReLU}\left(W_{\text{in}}^{(l)} \vec{a}^{(l-1)}\right)$$

The activations of the $i^{\text{th}}$ neuron is given by

$$n_i^{(l)} = \text{ReLU}\left((W_{\text{in}}^{(l)})_i \, \vec{a}^{(l-1)}\right)$$

where $(W_{\text{in}}^{(l)})_i$ the row vector of weights that feeds into neuron $i$.

The $j^{\text{th}}$ dimension of the output MLP can be rewritten as

$$\text{MLP}_j^{(l)}\left(\vec{a}^{(l-1)}\right) = \sum_i (W_{\text{out}}^{(l)})_{ji} \text{ReLU}\left((W_{\text{in}}^{(l)})_i \, \vec{a}^{(l-1)}\right)$$

We want to interpret this as an integral by doing something like:

$$\text{MLP}_j^{(l)}\left(\vec{a}^{(l-1)}\right) = \sum_i w_i f(\vec{a}^{(l-1)}; \xi_i)$$

$$\approx Z \int_{\xi_0}^{\xi_n} f(\vec{a}^{(l-1)}; \xi) \, \mathrm{d}\xi$$

$$= F(\vec{a}^{(l-1)})$$

Assume without loss of generality that $\xi_i$ is one-dimensional and $\xi_i < \xi_{i+1}$ for all $\xi$.

Note that we might have $F(\vec{a}^{(l-1)}) = \text{MLP}_j^{(l)}\left(\vec{a}^{(l-1)}\right)$, which would make this trivial.

What does it mean for this fact to be nontrivial?

We need:

1. A "locally one-dimensional" neuron-indexed variable of integration
2. Analytically described $f(a, \xi_i)$
3. $w_i$ should be approximately linear in $(\xi_{i+1} - \xi_{i-1})/2$
4. $f(\vec{a}^{(l-1)}; \xi_i) - f(\vec{a}^{(l-1)}; \xi_{i-1})$ is uniformly small over $\vec{a}^{(l-1)}$s (that is, $f$ is "continuous" in $\xi$ for all $a + b$).
5. We can analytically evaluate the integral

$$\int_{\xi_0}^{\xi_n} f(\vec{a}^{(l-1)}; \xi) \, \mathrm{d}\xi = F(\vec{a}^{(l-1)})$$

   independent of $\vec{a}^{(l-1)}$
6. We can bound the error of the numerical approximation of f at each $\xi_i$, independent of a. (e.g. lipschitz constant $\times$ size of box)

Ordinarily, we might check this approximation by empirically validating that

$$\left|\text{MLP}_j^{(l)}\left(\vec{a}^{(l-1)}\right) - F(\vec{a}^{(l-1)})\right| < \varepsilon$$

for some small $\varepsilon$, over all $\vec{a}^{(l-1)}$s

The reason this is non-trivial is you might be able to evaluate the error in the integral uniformly across all possible $\vec{a}^{(l-1)}$s.

## L    FURTHER WORK

There are several hurdles to replicating this approach to interpreting other neural networks. It is highly labour intensive, and requires extensive mathematical exploration. Thus, in order for the approach to have any practical use, we need to develop automated tools to make these approximations and interpretations.

For example, we may want to use the first few terms of the Fourier expansion (or other low-rank approximations) to approximate the action of various layers in a neural network, and then combine those to get algebraic expressions for certain neuron outputs of interest. Such algebraic expressions will natural admit phenomena like the numerical integration we described above. This sort of method may be particularly fruitful on problems which Fourier transforms play a large role, such as signal processing and solutions to partial differential equations.

## M    FUTURE TECHNICAL WORK

To complete the technical work laid out in section 7, we must accomplish two tasks which we discuss in this appendix section: constructing a parameterisation of the MLP which is checkable in less than $\mathcal{O}(p \cdot d_{mlp})$ time, and more generally constructing a parameterisation of the entire 'pizza' model that is checkable in time that is linear in the number of parameters; and establishing a bound on the error in the model's logits that does not neglect any terms.

### M.1    LINEAR PARAMETERISATION

Constructing a parameterisation of the model which is checkable in less than $\mathcal{O}(p \cdot d_{mlp})$ time is a relatively straightforward task, given the interpretation in the body of the paper. We expect that the parameters are:

- A choice of $n_{\text{freq}}$ frequencies $k_i$.
- A splitting of the neurons into groups by frequency, and an ordering of the neurons within each group.
- An assignment of widths $w_i$ to each neuron, and an assignment of angles $\phi_i$ to each neuron.
- An assignment of orthogonal planes into which each frequency is embedded by the embedding matrix, and by the unembedding matrix.
- Rotations and scaling of the low-rank subset of the hidden model dimension for each of the O and V matrices.

### M.2    BOUNDING THE ERROR OF THE MLP

To bound the error of our interpretation of the MLP precisely, we'd need to include a bound on the primary frequency contribution of the identity component (which integrates to 0 symbolically), and include bounds on the residual components – OVE on $x$ and $y$, the MLP bias, and the embed of $y$, as inputs to ReLU; and UOVE on $x$ and $y$ and UE on $y$ as output logits.

We could decompose every matrix in our model as a sum of the corresponding matrix from our parameterized model and a noise term. Expanding out the resulting expression for the logits (and expanding $|x + \varepsilon|$ as $|x| + (|x + \varepsilon| - |x|)$), we will have an expression which is at top-level a sum of our parameterized model result and a noise term which is expressed recursively as the difference between the actual model and the parameterized model. We can then ask two questions:

1. What worst-case bounds can we prove on the error terms at various complexities?
2. What are the empirical worst-case bounds on the relevant error terms?

