# OpenReview forum: "Modular addition without black-boxes: Compressing explanations of MLPs that compute numerical integration"
_ICLR.cc/2025/Conference — Submitted to ICLR 2025_

### Official Review · Reviewer_Scwe · 2024-10-31

**Soundness:** 3
**Presentation:** 2
**Contribution:** 3
**Rating:** 6
**Confidence:** 2

**Summary:**

The authors address the challenge of interpretibility of MLP layers in transformer models. Focusing on toy models trained for modular addition, the authors go beyond “approximate explanation” and propose to form a “complete interpretation” through a quadrature scheme, where neurons can be understood as approximating areas under curves in trigonometric functions. The method allows them to bound the MLP in total time linear in the neuron numbers.

**Strengths:**

- The authors provides a depth analysis, with a formal definition of “complete interpretation”, a clear problem setting, and an evaluation on a simplified toy model, which enables a focused and mathematically rigorous analysis.

- To evaluate the practical usage of the numerical integration interpretation, the authors validate the methods by establishing non-vacuous bounds for the MLP outputs in linear time.

**Weaknesses:**

- The paper is in general hard to follow for a broader audience, especially for those unfamiliar with topics such as mechanistic interpretability, quadrature themes etc.

- A discussion on the scalability beyond the toy models is missing. While a simple toy model allows for a focus analysis, a discussion on scaling to more complex real-word tasks is beneficial.

- As the author mentioned that there are couple of previous studies on the interpretability of MLPs, a comparative study with existing methods would be beneficial to fully understand the advantage of the proposed method.

**Questions:**

- Is it possible to extend the proposed framework to MLP layers in a deeper transformers?

- The study seems only focuses on understanding MLPs within a constrained setting where the MLP approximates trigonometric functions, is it possible to extend to non-trigonometric functions?

- I am still confused on the definition of “complete interpretation”, are there any evaluation metrics?

---

> ### Author Response · Authors · 2024-11-28
>
> Thank you for reviewing our paper, and for providing constructive feedback.  Per your general feedback, we've posted an updated manuscript: updated section 1 with more problem motivation, comparison with related work in section 2, and the remaining paper with notation changes and additional sentences and labels to make it easier to follow.
>
> > Is it possible to extend the proposed framework to MLP layers in a deeper transformers?
>
> Yes. We can restrict the analysis to an MLP layer within a deeper transformer and get some analytic expression for what the layer is doing. However, piecing together what the network does as whole would be more difficult, since error bounds can explode over multiple layers of the network. Please refer to the general response (the section on applications of our methods).
>
> > The study seems only focuses on understanding MLPs within a constrained setting where the MLP approximates trigonometric functions, is it possible to extend to non-trigonometric functions?
>
> Yes. As long as we have an analytic expression for the MLP weights, we can potentially use the idea of numerical integration to analyse the layer. Please refer to the general response (the section on applications of our methods).
>
> > I am still confused on the definition of “complete interpretation”, are there any evaluation metrics?
>
> We have removed the framing of “complete interpretation” and instead rewritten section 3 to provide a clearer connection with [1], upon which we build our validation procedure. The evaluation metric is the computational complexity needed to check the interpretation. The lower the complexity, the better we've compressed the model, and hence the better our interpretation. We believe that a computation linear in the number of parameters is the best one can achieve here, since we would have to look through all the weights only once.
>
> Please let us know if there are any other clarifications we can provide or improvements we can make!
>
> [1] Gross et al. Compact Proofs of Model Performance via Mechanistic Interpretability. (2024)

---

> ### Comment · Reviewer_Scwe · 2024-11-30
>
> Thank you for your response and for addressing some of my concerns. I think the revisions improve the clarity of your paper. The clarification on the challenges in extending the framework to deeper transformers, and the explainations on potential applications are beneficial. However, I feel these updates do not fully address the concern regarding the scalabilty and practical applications in more complex settings. Therefore, I will maintain my original score.

---

> ### Author Response · Authors · 2024-12-01
>
> Thank you for your response. We want to note that we're the first paper to develop an approach to interpreting densely-connected MLP layers. While we cannot make changes right now, we plan on adding a section clearly detailing the structure of the infinite-width argument that we make in this paper, which is applicable to other settings.
>
> Consider a nonlinear layer $\text{MLP}\left(a\right) = W_{out} \text{ReLU} \big (W_{in} a \big)$, where activation $n_i$ of neuron $i$ is given by $\text{ReLU}\big ((W_{in})_{i} ~ a \big)$. Our goal is to find an approximation $F(a)$ of each output $\text{MLP}_j\left(a\right)$ such that we can bound $ |{\text{MLP}_j\left(a\right)  - F(a)}| < \varepsilon$ without computing $\text{MLP}_j\left(a\right)$ for each $a$. While $\text{MLP}\left(a\right)$  is a discrete sum over finitely many neurons, we are trying to find an analytic approximation $F(a)$.
>
> The infinite-width lens argument is constructed by an approximation of the discrete sum of the MLP output as follows: $\sum_i w_i f(a; \xi_i) \approx Z \int{\xi_0}^{\xi_n} f(a; \xi) ,\mathrm{d}\xi = F(a)$.
>
> where
> -  $\xi_i$ is a ``locally one-dimensional" neuron-indexed variable of integration
> -  $w_i$ is  approximately linear in $(\xi_{i+1} - \xi_{i-1})/2$
> - $f(a; \xi_i) - f(a; \xi_{i-1})$ is uniformly small over all values of $a$, i.e. $f$ ``continuous" in $\xi$
>
> If we are able to find values satisfying the constraints, then we can analytically evaluate the integral $\int_{\xi_0}^{\xi_n} f(a; \xi) ,\mathrm{d}\xi = F(a)$ independent of $a$, allowing us to bound the error of the numerical approximation of $f$ at each $\xi_i$, independent of $a$, for example via lipschitz constant $\times$ size of box.
>
> Each mechanistic interpretability case study is a fair bit of human labor, and deriving these values will not be trivial. Moreover, without empirically checking, we cannot say how good a given approximation is. However, our approaches to error-bounding, and constructing the infinite-width / numerical integration approximation are sound, and general purpose solutions to interpreting MLPs.

---

### Official Review · Reviewer_Si51 · 2024-11-01

**Soundness:** 2
**Presentation:** 2
**Contribution:** 1
**Rating:** 3
**Confidence:** 5

**Summary:**

This paper examines how 2-layer ReLU MLPs implement the trigonometric identities required by the 'Pizza' algorithm (previously described in earlier work) within a one-layer modified transformer architecture. The authors found that the MLP approximate a trigonometric integration by summing over its hidden neurons, they also provide non-vacuous bounds on the output of the MLP on all inputs in time linear in the parameters.

**Strengths:**

1. Simple setting with a clear explanation of their setting
2. Offered empirical check of their theoretical results

**Weaknesses:**

The work only focuses on explaining details of a component from the previously proposed Pizza Algorithm (Zhong et al., 2023), specifically how MLPs sum periodic features generated before activation functions. Furthermore, the simplified model (lines 125-126) is nearly identical to the fully solved model presented in [1], with ReLU activation replacing the quadratic activation. Since exact finite summation for ReLU remains an open problem, the authors instead approximate the sum using integral methods and argue that neurons effectively approximate this integral.

While I appreciate the authors work on explaining the role of secondary frequencies in Section 5, it represents only a minor contribution. Overall, this paper's contribution falls below the standard expected at ICLR-level conferences.

[1] A. Gromov, Grokking modular arithmetic, arXiv:2301.02679

**Questions:**

1. Have the authors investigated the effect of weight decay tuning on the clarity of their experimental results?
2. What factors influence the emergence of secondary frequencies in the model?

Line 023: "trignometric" typo in abstract
Line 992: Broken equation ref

---

> ### Author Response · Authors · 2024-11-28
>
> Thanks for taking the time to review our paper. We've posted an updated manuscript incorporating feedback.
>
> We want to start by addressing your concern that our contribution is not significant. We appreciate this note, and have modified sections 1, 2, 3, 7 to frame and describe our contribution more clearly.
>
> 1. We build upon [1] which is a fully formal approach to mechanistic explanation of models. In [1], the authors only interpret one layer transformers with no MLP layers, trained on a simpler task of computing the max of k integers. Our work applies the formalization to a new, harder case study.
> 2. Mechanistic interpretability is generally intensive labor, and formal mechanistic interpretability is even more intensive!
> 3. To our knowledge, we are the first to interpret MLP layers rigorously. We think this is a significant direction of exploration. By interpreting feature-maps, we are able to uncover a completely novel algorithmic description of the model. This is important for applications of mech interp geared at high stakes applications like anomaly detection and guarantees, but also for lower-stakes applications like model distillation and model steering. We consider our work as framing that feature-maps are important to interpret, and then concretely showing that interpreting them adds insight even to the models that are considered best understood in mech interp.
>
>
> > Have the authors investigated the effect of weight decay tuning on the clarity of their experimental results?
>
> We use large weight decay when training the network so that the weights have smaller L2-norm, which gives rise to a more compact model. This is demonstrated in previous papers about grokking. However, our paper falls in the paradigm od post-hoc mechanistic interpretability, which we have clarified in section 3. Thus, we take as given that the networks are trained so that they perform perfectly on the dataset.
>
> > What factors influence the emergence of secondary frequencies in the model?
>
> As stated in section 5, secondary frequencies improve the effectiveness of the model by improving its performance when cos(k/2 (a-b)) \approx 0. It can be interesting in future work to investigate the development of secondary frequencies over the training phase, and also the optimal strength of the secondary frequency with respect to the loss function.
>
> Let us know if you have any more questions or suggestions! We're happy to incorporate any feedback you have on our work.
>
> [1] Gross et al. Compact Proofs of Model Performance via Mechanistic Interpretability. (2024)

---

> ### Author Response · Authors · 2024-12-02
> **Polite Reminder to Reviewer Si51**
>
> Given that the discussion period ends in approximately 16 hours, we would greatly appreciate the reviewer's thoughts on our updates and replies. We welcome any additional questions or comments, and remain happy to address them to the best of our ability within the remaining time.

---

> > ### Comment · Reviewer_Si51 · 2024-12-03
> >
> > I appreciate the authors for their efforts in editing the paper and addressing my questions.
> > However, I remain unconvinced that the paper makes a sufficiently significant contribution to the mechanistic understanding of grokking. Therefore, I will maintain my current score.

---

> > > ### Author Response · Authors · 2024-12-03
> > > **Grokking?**
> > >
> > > Thank you for your continued engagement with our paper. We feel it's important to address a misunderstanding about our work's focus. Our contribution is not to the mechanistic understanding of grokking, but rather provides the first fully rigorous mechanistic analysis of MLP behavior in trained models.
> > >
> > > Let us express in more detail how our work advances beyond prior research:
> > >
> > > Gromov [1] proposed theoretical equations for MLPs but provided only minimal empirical validation that trained models implement these equations. Our work, in contrast, provides concrete rigorous evidence and bounds for how closely real MLPs approximate trigonometric computations using our interpretation. A crucial limitation of [1] is that equations (11), (13), and (16) assume a uniform distribution of frequencies across neurons and single-frequency behavior per neuron. While the IPR measurement in Section 4.3 of [1] captures Fourier basis sparsity, it doesn't validate either the uniform distribution of frequencies among neurons nor the claim of uncorrelated, random phases. Notably, while Appendix B of [1] claims their solution works for ReLU activations, it overlooks that standard training with ReLU typically learns an algorithm incompatible with their analysis. Given the high expressivity of MLP layers, even if networks with quadratic activations implement [1]'s algorithm, the evidence presented doesn't demonstrate this conclusively. We touch on these points in sections 2 and 3 of our paper.
> > >
> > > We should also mention that our proof in Appendix F of our paper extends trivially beyond ReLU to any non-negative activation function, including [1]'s squaring activation.  Simply removing ReLU from each equation preserves the proof's validity -- we've since updated our text to reflect this more general proof. We initially limited our discussion to ReLU since we haven't empirically verified that networks using other activation functions learn our proposed algorithm rather than [1]'s or alternative approaches.
> > >
> > > Nanda et al. [2] identified Fourier components before and after the MLP but didn't explain the mechanism of this transformation. Similarly, while Zhong et al. [3] described the MLP's functional output, they didn't provide a mechanistic explanation of how it achieves this transformation.
> > >
> > > Summarizing, our paper:
> > > - Provides the first rigorous analysis of *how* trained MLPs implement trigonometric computations in practice
> > > - Offering provable bounds on empirical divergence of MLP behavior
> > >
> > > We believe this represents a significant contribution to mechanistic interpretability of neural networks, particularly in understanding how to validate that theoretical predictions manifest in trained models, especially in the case of MLPs. This kind of detailed component analysis is essential for advancing reliable and transparent AI systems.
> > >
> > > We welcome any additional discussion about the contribution of our paper that may be possible before the deadline.
> > >
> > > [1] A. Gromov, Grokking modular arithmetic, arXiv:2301.02679
> > > [2] Nanda et al., Progress measures for grokking via mechanistic interpretability, arXiv:2301.05217
> > > [3] Zhong et al., The Clock and the Pizza: Two Stories in Mechanistic Explanation of Neural Networks, arXiv:2306.17844

---

### Official Review · Reviewer_1gJr · 2024-11-04

**Soundness:** 4
**Presentation:** 3
**Contribution:** 3
**Rating:** 8
**Confidence:** 4

**Summary:**

MLP layers in transformer models remain poorly understood in the mechanistic interpretability literature; even detailed analyses of toy models like modular addition transformers tend to treat them as black boxes that compute certain functions and verify their input-output behavior by exhaustive search over the input space. The authors propose to fully explain—by which they mean find a description that is linear in the number of parameters—the MLP's behavior in the modular addition task. They determine that the MLP is responsible for numerically computing an integral of a trigonometric identity that is useful for the modular addition task, and even prove non-vacuous bounds on the error of their description.

Edit: confidence increased 3->4 during rebuttals

**Strengths:**

* **Addressing open problems in mechanistic interpretability.** While modular addition is a well-studied problem, the MLP sublayer's behavior has mostly been treated as a black box. By explaining what the MLP is doing, the authors help move the study of this model beyond this black box treatment. While understanding one piece of a toy model may seem like a modest contribution, this is a major missing piece of an active area of research and a major step towards producing and verifying mathematical descriptions of MLP behavior.
* **Resolving the discrepancy between "clock" and "pizza" algorithms.** In section 5, the authors solve the mystery of why "pizza" algorithm logits resemble "clock" logits by explaining the role that secondary frequencies play. This is also an unresolved question in modular addition.
* **Balance of perspectives.** The authors are thorough about including empirical/computational results, mechanistic descriptions of the modular addition task, and proofs. They also extend results to 150 other transformer models (which differ in their random seed).

**Weaknesses:**

* **Limited scope.** Although the paper is thorough and addresses an open question in mechanistic interpretability, its concrete contribution is simply to explain one component of one model (albeit under multiple random seeds) on one toy problem. While this is a step forward, and the authors briefly describe the broader implications for non-algorithmic interpretability practitioners in Section 7, the paper's contribution is circumscribed (arguably modest) as written.
* **Bounds are crude.** As the authors acknowledge (354), the bounds they prove are quite loose, owing to an incomplete understanding of the model. Proving tighter bounds or arguing more convincingly for why tighter bounds are not possible would improve the paper.
* **Clarity.** The argumentation in some of the mathematically dense sections (especially Section 5) can be hard to follow.

**Questions:**

* You suggest that this approach could be useful for practitioners in other domains. What properties of the model/problem would make it more likely that this approach would yield a good explanation?
* You leave deriving tighter error bounds to future work. What avenues for proving these bounds are you considering?

---

> ### Author Response · Authors · 2024-11-28
>
> Thank you for your support of our paper, we appreciate it!
>
> > Limited scope
>
> We have clarified the contribution of this paper to be a prototype / first case study in compressing feature-maps. We believe that this is an important problem, and have updated our introduction.
>
> > Bounds are crude
>
> The standard we adopt for mechanistic explanation is formal proof. Prior work has only empirically developed this on transformers with no MLP layers. Thus, despite the crudeness of bounds, we think that getting non-vacuous bounds itself in a step forward.
>
>
> > You suggest that this approach could be useful for practitioners in other domains. What properties of the model/problem would make it more likely that this approach would yield a good explanation?
>
> When the model has a clear functional/mathematical form, this approach of numerical integration would work better, as we can use similar methods to guess the functional form implied by the network and bound the errors. As such, this can be useful when we look at neural networks used to solve mathematical problems (e.g. solving PDEs). However, it can also be used in other domains: if we spot some analytical expression in certain parts of the network, we could potentially convert this to a statement about the input / output space of the problem. Please refer to the general response for more details.
>
> > You leave deriving tighter error bounds to future work. What avenues for proving these bounds are you considering?
>
> One can use properties of quadrature schemes to derive tighter bounds. Currently, we are using the naive estimate of transforming a weighted sum to rectangles under a curve. There may be better methods from numerical analysis (Gaussian quadratures etc.) which give better bounds.
>
> We hope we've been able to address your concerns about the paper. Please let us know if there's any more information we can provide, or changes we can make!

---

> > ### Comment · Reviewer_1gJr · 2024-12-01
> >
> > Thank you for your edits. I agree that the paper is clarified by the current edits. Since I have already given the paper an 8, I will keep my score, but I will raise my confidence to a 4.

---

> > > ### Author Response · Authors · 2024-12-02
> > >
> > > Thank you!

---

### Official Review · Reviewer_MPRr · 2024-11-04

**Soundness:** 2
**Presentation:** 2
**Contribution:** 2
**Rating:** 5
**Confidence:** 4

**Summary:**

The paper studies the role of MLP layers in a constant-attention Transformer (i.e. ReLU-MLP network) trained on modular addition. They write the overall operation of the network as $embedding \\rightarrow ReLU \\rightarrow fully-connected$ and write the trigonometric functions corresponding to these components. They show that the sum over same-frequency neurons implements a sum over random phases, which can thought of as approximating an integral. They provide empirical evidence for their claims and show that their method enables the computation of non-trivial error bounds on the outputs.

**Strengths:**

- The analysis of secondary frequencies is new.

- The results are presented in a quite candid fashion, with the limitations/discrepancies of the analysis freely discussed, along with potential explanations.

- The code used to produce the results is provided in the form of a link to a Colab Notebook.

**Weaknesses:**

My main concerns lie in regards to the novelty and soundness of this work.

- The authors only consider Transformers with constant attention. This makes the network a ReLU-MLP and not a Transformer. This should be made clear in the paper. If the objective is to study Transformers, then non-constant attention should also be examined.

- Interpretability for MLPs trained on modular addition has already been extensively studied in literature [1,2]. Although these prior works focused on MLPs quadratic activations, many of the qualitative conclusions are similar to the ReLU case studied in this work. The authors should present a detailed comparison to these works and highlight the novelty. (Specifically, what are the insights from the current work that are lacking upon merging the findings of [1,2,3].)

- As stated, it is difficult to draw conclusions from the error bounds -- especially the part with splitting ReLU into identity and absolute value components. This part would benefit from further justification as well as clarity of notation (e.g. lines 309-310).

**Questions:**

- Are all the experiments conducted with constant head Transformers? I couldn't find this detail in the Colab notebook since the checkpoints are directly imported for analysis.

- According to [3], networks can learn either "clock" or "pizza" algorithm, depending on the details of training. How do the authors ensure that the learned algorithm is always "pizza"?

- Are Figures 2,5 empirical results or just theoretical visualizations?

- Lines 299-303: The omission of x/2 term seems rather hand-wavy. How does one see that computing the bounds after ignoring this term is a reasonable thing to do? It is unclear to me even after referring to Appendix K.

- Does Table 1 imply that the contribution from $\\varepsilon_\\phi$ is very large? This seems unlikely since $\\psi_i - 2\\phi_i$ are quite small in Figure 3.

- Line 349: Does "brute force" simply mean empirical?

- Line 373-377: Zhong et al. [3] do not claim that logits are always of the "pizza" form. They claimed that either "clock" or "pizza" algorithm can be found, depending on the hyperparameters.

- Lines 395-397: I am puzzled by the $R^2$ results. The paper sets out to explain the "pizza" algorithm. However, the logits seem to align better with the "clock" algorithm.

- Lines 401-403: This paragraph claims that the secondary frequencies are important. However, Figure 4 shows that the top Fourier component explains more than 99% of the variance in all of the neurons. How does one reconcile these two results?

[1] Gromov, Grokking modular arithmetic (2023)

[2] Doshi et al., To grok or not to grok: Disentangling generalization and memorization on corrupted corrupted algorithmic datasets (2024)

[3] Zhong et al., The Clock and the Pizza: Two Stories in Mechanistic Explanation of Neural Networks (2023)

---

> ### Author Response · Authors · 2024-11-28
>
> Thank you very much for your time and feedback! We have uploaded an updated manuscript incorporating recommended changes.
>
> > The authors only consider Transformers with constant attention.
>
> We’re following the terminology of Zhong et al, since we’re extending their interpretation. However, we've made the scope of our work clearer, and emphasized that we are studying the ReLU feature-map only.
>
> > Interpretability for MLPs trained on modular addition has already been extensively studied in literature.
>
> We've modified section 2 to start with a description of prior work on analyzing modular addition models. We've clearly framed that our goal is to compress feature-maps, which prior work does not do. Moreover, we clarify that we worked on the modular addition model since they are some of the best understood models in literature. Thus, by finding more insight from analyzing the MLP, we  demonstrate that the problem we're interested in is important.
>
> > Are all the experiments conducted with constant head Transformers? I couldn't find this detail in the Colab notebook since the checkpoints are directly imported for analysis.
>
> Yes. (i.e. there is no attention that is dependent on the inputs.) We trained the models with a separate codebase, so it is not shown in the Colab notebook.
>
> > According to [3], networks can learn either "clock" or "pizza" algorithm, depending on the details of training. How do the authors ensure that the learned algorithm is always "pizza"?
>
> Our preliminary analysis of the clock model (which is not included in the write-up) suggest that the primary difference between the "clock" and the "pizza" model is where the transition from $\cos a + \cos b \to \cos a \cos b$ happens.  In the pizza model, we have (by construction) addition of embeds pre-ReLU, thus resulting in pre-activations which have significant coefficients in the $\cos a \cos b$ and $\sin a \sin b$ Fourier components.
> By contrast, the "clock" models seem to do some of this multiplication in the attention layer (as recognized by Nanda et al), but no neuron's pre-activations contain Fourier components of both $\cos a$ (or $\sin a$) and $\cos b$ (or $\sin b$).  These components are then added post-ReLU to get a dependence on $a+b$.
> Thus the "clock" algorithm (using attention to get $\cos a \cos b$ or $\sin a \sin b$ components for unembed, and also forwarding only either a or b to the MLP) is impossible to implement in the architecture we use (constant attention).
> Regarding the consistency of learning the "pizza" algorithm, the results from 150 random seeds (Appendix C in our paper), most of the models agree with the formula for the weights that we suggested. One might consider this to define the “pizza” algorithm.
>
>
> > Are Figures 2,5 empirical results or just theoretical visualizations?
>
> They are empirical results coming from the trained models and our calculations as detailed in section 4.1.
>
> > Lines 299-303: The omission of x/2 term seems rather hand-wavy. How does one see that computing the bounds after ignoring this term is a reasonable thing to do? It is unclear to me even after referring to Appendix K.
>
> This is a quirk with our error metric. We are considering relative error, i.e.
> $\frac{error bound}{theoretical value}$. For the x/2 term, the theoretical value of the integral is 0, hence relative error would not make sense. Thus, we focused on the $|x|/2$ term (with non-zero theoretical value) instead, to demonstrate how our method works. It is possible to compute bounds for ReLU as a whole (see Appendix I).
> Also, the $x/2$ term is in a sense “simpler” than the $|x|/2 term$, because it is just a linear layer. Indeed, we can find a matrix $A \in \mathbb{R}^{p \times p}$ such that the logit contribution from this part of the network is $A[a,:] + A[b,:]$, for inputs $(a, b)$. Thus, this part of the network can be seen as already interpreted in time linear in the number of parameters.

---

> > ### Author Response · Authors · 2024-11-28
> >
> > > Does Table 1 imply that the contribution from $\epsilon_{\phi}$ is very large? This seems unlikely since $\psi_i - 2\phi$ are quite small in Figure 3.
> >
> > The second row of Table 1 consists of the contribution from \eps{\phi} and the error bound we get by our “compact” argument. The first row is the actual error computed by brute force over all possible inputs. Indeed, the error bound contribution from the angles is much smaller than that from the quadrature scheme (see Appendix I for a detailed breakdown of our error bound).
> >
> > > Line 349: Does "brute force" simply mean empirical?
> >
> > Yes. This means evaluating the model over all possible inputs. We use "brute force" following prior work, and to indicate the infeasibility of scaling up this approach.
> >
> > > Line 373-377: Zhong et al. [3] do not claim that logits are always of the "pizza" form. They claimed that either "clock" or "pizza" algorithm can be found, depending on the hyperparameters.
> >
> > This was a mistake in communication. We changed the wording to express that the “pizza” algorithm was suggested in [3].
> >
> > > Lines 395-397: I am puzzled by the $R^2$ results. The paper sets out to explain the "pizza" algorithm. However, the logits seem to align better with the "clock" algorithm.
> >
> > The point here is that the “pizza” algorithm is implemented by the $|x|/2$ component of the ReLU, and the presence of the secondary frequencies adjusts the logits to look more like the “clock” algorithm.
> >
> > > Lines 401-403: This paragraph claims that the secondary frequencies are important. However, Figure 4 shows that the top Fourier component explains more than 99% of the variance in all of the neurons. How does one reconcile these two results?
> >
> > (1) The effect of secondary frequencies is not very big (e.g. the change in $R^2$ from 0.85 to 0.99 is not very large). (2) Even if the top Fourier component explains 99% of the variance, the second Fourier component can explain 1% of the variance i.e. 10% of the standard deviation, which is not small. One can see this as the primary frequency implementing the main part of the algorithm, with the secondary frequency playing a supporting role (and helping to improve the model).
> >
> > > As stated, it is difficult to draw conclusions from the error bounds -- especially the part with splitting ReLU into identity and absolute value components. This part would benefit from further justification as well as clarity of notation (e.g. lines 309-310)
> >
> > We've added further description of the verification procedure in section 3. We have also clarified the notation in section 5, which is now lines 357-359.
> >
> >
> >
> > We hope we’ve been able to address the issues and questions you've asked. Please let us know of instances where we can improve, we value the ways in which we are able to improve our paper from your feedback.

---

> ### Author Response · Authors · 2024-12-02
> **Polite Reminder to Reviewer MPRr**
>
> Given that the discussion period ends in approximately 16 hours, we would greatly appreciate the reviewer's thoughts on our updates and replies. We welcome any additional questions or comments, and remain happy to address them to the best of our ability within the remaining time.

---

> ### Comment · Reviewer_MPRr · 2024-12-03
>
> I thank the authors for the detailed responses to my questions and concerns.
>
> The answers and the manuscript-edits address some of my concerns about clarity and soundness. However, I continue to have reservations about the significance of this work towards understanding Transformers trained on modular arithmetic and the field of mechanistic interpretability.
>
> Consequently, I will increase my score but remain below acceptance.

---

### Author Response · Authors · 2024-11-28

Thank you to the reviewers for your detailed engagement with our paper, and thank you for patience while we incorporated the feedback. We have uploaded a new draft of our paper. Here, we will take the opportunity to respond to some feedback to common to reviewers.

**Framing of the paper:** Mechanistic interpretations currently compress activations into features, but do not compress the feature-maps itself. Whereas linear layers are easy to compress, nonlinear feature maps like MLP layers are quite expressive and challenging to compress. To our knowledge, we are the first to rigorously compress an MLP.

**Prototyping using the modular addition models:** We worked on the modular addition models given that they are some of the best interpreted models. However, even in these models, the MLP layer is treated like a black box. Our goal was to compress the MLP, and demonstrate that this adds additional insight about the model that was previously not there.

**Comparison with prior work:** We have updated section 2 to start with a discussion of prior interpretations.
- Originally, [1] studied a one-layer transformer model. They found low-rank features to describe all components of the model, and analyzed the feature-map of the final linear layer. While they generated a human-intuitive algorithm of how the model works, they did not explain how the MLP layers compute logits that fit with the form required by their algorithm. Despite the tremendous progress in analyzing the non-MLP layers of the model, they still treat the ReLU MLP as a black-box.
- [2] extended the analysis to a family of architectures parameterized by attention rate. The architecture from [1] corresponds to attention rate 0, while attention rate 1 corresponds to a ReLU MLP-only model, i.e. a transformer with constant attention. Depending on attention rate, they showed that models may learn the ``clock'' or ``pizza'' algorithm. They exhaustively enumerated inputs to MLP-only model and found a description of the feature-map. However, exhaustive enumeration is not feasible for larger input sizes, and does not constitute an insight-rich explanation. Thus, their approximation of the feature-map is equal to the feature-map itself, failing to provide any compression / insight.
- [3] considered a  cleaner version of the MLP-only architecture considered by \citet{zhong2023clock}, and used quadratic activations instead of ReLU activations. They presented a formula corresponding to a compressed feature-map. However, they did not present sufficient evidence to show that the trained models follow the formula as suggested. They only showed that the weights are roughly “single-frequency”, i.e. they are well approximated by the largest Fourier component. Establishing that the model in fact uses the stated algorithm as opposed to a different algorithm would require significantly more validation.
- In our work, we analyze the ReLU MLP to show how it computes the functions described in prior work, and use rigorous evidence and proofs to validate our interpretation. We apply the infinite width lens and obtain an analytic expression, uncovering a completely different interpretation of the ReLU model, where we see ReLU as evaluating numerical integration. Finally, we demonstrate that using this interpretation gives us an efficiently computable bound such that we don’t have to exhaustively enumerate all possible inputs to describe the numerical integration function, i.e. we find a meaningful compression of ReLU.

**Applications of our methods:**
- Our general methodology can be applied if we can find an explicit functional form taken by some componentr of a NN. Then, we can consider the difference between the model weights and the weights suggested by the functional form, and bound the error like how we did in the paper. In deeper networks, it would be difficult to produce a coherent and rigorous error bound for the whole network due to how error propagates, but we can certainly focus our analysis on specific layers of the network, or specific data distributions.
- Numerical integration, specifically, can be applied if we know the form of the model weights. E.g. consider a general ReLU-MLP layer $y_2 = \sum_i \mathrm{ReLU}(y_{1,i}) w_i$. If we can express $y_{1,i} = f(x,\phi_i)$ where $x$ is the input, then we could investigate the possibility of $y_2 = \sum_i \mathrm{ReLU}(f(x,\phi_i) w_i \approx \int \mathrm{ReLU}(f(x,\phi))\,\mathrm{d}\phi$. This turns a finite sum expression into an integral expression which can be evaluated analytically (if we can find the functional form). An analytic expression adds a lot to the interpretability of the network.
[Maybe come up with some similar expression when y_{1,i} is low rank]

[1] Nanda et al., Progress measures for grokking via mechanistic interpretability (2023)
[2] Zhong et al., The Clock and the Pizza: Two Stories in Mechanistic Explanation of Neural Networks (2023)
[3] Gromov, Grokking modular arithmetic (2023)

---

### Meta-Review · Area_Chair_hLeV · 2024-12-19

**Metareview:**

This paper tackles the interpretability of MLP layers in transformer models by studying the role of MLP layers in a constant-attention Transformer (i.e., ReLU-MLP network) trained on modular addition. The paper was reviewed by four reviewers (scores 3+5+6+8) who give the paper an average score of 5.5. The reviewers were divided on the strengths and weaknesses of this paper. The reviewers gave the paper credit for addressing an interesting problem and resolving connections between existing algorithms. At the same time, the reviewer consensus was that the paper has problems with limited scope (only considering constant-attention Transformers, which is limiting), limited practical benefits in the analysis, and clarity issues and confusing presentation across the paper. The discussion phase was helpful but understandably did not resolve all the concerns.

Summarising the views from the reviews: Even if the analysis and experimentation in the paper are interesting, the limited setting considered under the constant-attention Transformer leaves the contributions short and the paper does not quite meet the bar for acceptance.

**Additional Comments On Reviewer Discussion:**

This paper was subject to active discussion and both the authors and reviewers were active. The average score increased from 5.0 -> 5.5 during the discussion.

---

### Decision · Program_Chairs · 2025-01-22

Reject